# *In situ* measurements of soil and plant water isotopes: A review of approaches, practical considerations and a vision for the future

Matthias Beyer[1], Kathrin Kühnhammer[1], Maren Dubbert[2]

[1] Institute for Geoecology, Technische Universität Braunschweig, Langer Kamp 19c, 38106 Braunschweig, Germany & German Federal Institute for Geosciences and Natural Resources (BGR), 30655 Hannover, Germany
[2] Ecosystem Physiology, University Freiburg, Georges-Köhler-Allee 53, 79110 Freiburg

*Correspondence to*: Matthias Beyer (matthias.beyer@tu-bs.de)

**Glossary**

**bag equilibration method:** A methodology to determine the isotope composition of water/soil/plant samples which is based on collecting the measurand in air-tight bags which are then filled with a dry gas. Subsequently, the equilibrated vapor is directed to a laser spectrometer and measured. We count this method to 'destructive' sampling here as material needs to be removed from its origin in order to by analyzed.

**CG model**: Craig and Gordon model for isotopic fractionation during evaporation of open water bodies. Widely applied to determine the isotope composition of soil evaporation.

**cryogenic (vacuum) extraction:** Up to date most widely used method for extracting water from soil and plant samples for subsequent analysis of the isotope composition. In this method the water is extracted by heating the sample thus completely evaporating contained water and collection of the evaporate in a cold-trap.

**destructive sampling:** Any sampling activity where the material/substrate to be measured is removed from its original place (e.g. collection in vials, bags, etc.).

**gas permeable membrane:** Tubular micro-porous membrane material that is permeable for water vapor and allows the air inside to isotopically equilibrate with the surrounding of the membrane.

**gas permeable membrane probes:** Commercially available or custom-made probes employing gas permeable membranes

*in situ* **measurement:** A direct measurement of a variable in its original place (e.g. soil water isotopes). This term also applies to laboratory experiments using pre-built soil columns.

**IRIS:** Isotope-ratio infrared spectroscopy.

**IRMS:** Isotope-ratio mass spectroscopy.

**(liquid-vapor) isotopic equilibrium/equilibrium fractionation:** Difference (or fractionation) of the isotope composition between a liquid and a vapor phase that establishes in closed systems. This difference (or equilibrium fractionation factor) can be calculated via empirical equations that are dependent on temperature.

**isotope [or isotopic] composition [or signature]**: Term used to refer to the proportions of different isotopes in a sample. Not referring to a specific value (see $\delta$ [or delta] value). We use the term isotope composition throughout this manuscript.

**isotope ratio:** The ratio of the number of atoms of two isotopes in a material, herein the ratio of $^{18}O$ to $^{16}O$ or $^2H$ to $^1H$.

**$\delta$ [or delta] value, $\delta^2H$, $\delta^{18}O$:** Hydrogen and oxygen stable isotope ratio in delta notation; per definition it is a mathematical manipulation of a ratio of isotope ratios (the ratio of the sample compared to the ratio of an international standard) and refers to a specific value.

**$\delta_{vap}$, $\delta_{liq}$:** $\delta$ value of vapor and liquid phase.

**$\delta_E$, $\delta_T$:** $\delta$ value of evaporation and transpiration.

**laser spectrometer**: Measurement instrument using IRIS.

**kinetic fractionation**: Additional (to equilibrium fractionation) isotope fractionation in open systems, where air is not saturated and evaporated water vapor moves away from liquid water.

**open-split:** T-piece in front of the analyzer with tube attached to remove excess air in the throughflow system.

**soil water, pore water:** Soil water is a term used by soil scientists and refers to all water contained within the soil matrix (mobile and tightly bound water). Pore water is a term more commonly used by hydrogeologists; it accounts also for water contained in cracks/fissures, etc. which is per definition not soil water. We stick to the term 'soil water' here (unless the cited paper uses the term 'pore water' explicitly) as most studies were carried out in soils.

**vadose zone:** Unsaturated soil - zone between groundwater table and soil surface.

**water pools:** Different fractions of water withing a common source (e.g. mobile vs. immobile soil water).

**water sources:** The compartments where a plant potentially obtains its water from (e.g. soil water, groundwater, stream water).

**water vapor concentration:** Water vapor molecules in the sample air as determined by the laser spectrometer, unit ppmv.

**Abstract.** The number of ecohydrological studies involving water stable isotope measurements has been increasing steadily due to technological (e.g. field deployable laser spectroscopy and cheaper instruments) and methodological (i.e. tracer approaches or improvements in root water uptake models) advances in recent years. This enables researchers from a broad scientific background to incorporate water isotope-based methods into their studies.

Several isotope effects are currently not fully understood but might be essential when investigating root water uptake depths of vegetation and separating isotope processes in the soil-vegetation-atmosphere continuum. Different viewpoints exist on i) extraction methods for soil and plant water and methodological artefacts potentially introduced by them; ii) the pools of water (mobile vs. immobile) measured with those methods and iii) spatial variability and temporal dynamics of the water isotope composition of different compartments in terrestrial ecosystems.

*In situ* methods have been proposed as an innovative and necessary way to address these issues and are required in order to disentangle isotope effects and take them into account when studying root water uptake depths of plants and for studying soil-plant-atmosphere interaction based on water stable isotopes. Herein, we review the current status of *in situ* measurements of water stable isotopes in soils and plants, point out current issues and highlight potential for future research. Moreover, we put a strong focus and incorporate practical aspects into this review in order to provide a guideline for researchers with limited

previous experience to *in situ* methods. We also include a section on opportunities of incorporating data obtained with described *in situ* methods into existing isotope-enabled ecohydrological models and provide examples illustrating potential benefits of doing so. Finally, we propose an integrated methodology for measuring both soil and plant water isotopes *in situ* when carrying out studies at the soil-vegetation-atmosphere continuum. Several authors have shown that reliable data can be generated in the field using *in situ* methods for measuring the soil water isotope composition. For transpiration, reliable

methods also exist but are not common in ecohydrological field studies due to the required effort. Little attention has been payed to *in situ* xylem water isotope measurements. Research needs to focus on improving and further developing those methods.

There is a need for a consistent and combined (soils and plants) methodology for ecohydrological studies. Such systems should be designed and adapted to the environment to be studied. We further conclude that many studies currently might not rely on

*in situ* methods extensively because of the technical difficulty and existing methodological uncertainties. Future research needs to aim on developing a simplified approach that provides a reasonable trade-off between practicability and precision/accuracy.

# 1 Introduction

Since the presentation of the heavily debated 'two water worlds hypothesis' (McDonnell, 2014) the attention of many
ecohydrologists – especially those working with water isotopes – has been focussing on what was termed as 'ecohydrological separation'. In the original hypothesis, the authors claim that based on the studies of Brooks et al. (2010) and Goldsmith et al. (2012) plants in some watersheds prefer water which is 'more difficult' for them to access (i.e. soil water with relatively higher matric potential) over 'easier' accessible water sources (i.e. soil water with low matric potential that eventually becomes stream water).

The discussion remains controversial, with a number of criticism. Sprenger et al. (2016), for instance, offer a simple and logic explanation for 'ecohydrological separation': *"... subsequent mixing of the evaporated soil water with nonfractionated precipitation water could explain the differences in the isotopic signal of water in the top soil and in the xylem of plants on the one hand and groundwater and streamwater on the other hand"* (refer to Fig. 8 in Sprenger et al., 2016). Hence, the authors question "…if ecohydrological separation is actually taking part or if instead the soil water undergoes isotopic changes over
space (e.g., depth) and time (e.g., seasonality) leading to distinct isotopic signals between the top soil and subsoil, which will directly affect the isotopic signal of the root water." (Sprenger et al., 2016). Furthermore, plant physiological (rooting depth, water potential of plants) and aspects such as nutrient availability or the interplay between water demand vs. water availability were completely neglected in the theory (which the authors themselves admit, McDonnell, 2014). Especially the latter aspects have been omitted in many studies (partially this might be because many of those were conducted by hydrologists, not plant
experts). Plants might not want the 'easily accessible' water, for instance if this water is poor in dissolved oxygen or nutrients, and therefore use the 'less available' water preferentially. An example for this might be tropical catchments where soils are often nutrient-poor, but stream or fresh rainwater contains the majority of nutrients. Recently, Dubbert and Werner (2019) stated that isotopic differences between soil, plant and groundwater can be fully explained by spatio-temporal dynamics and that based on a pool-weighted approach, the effect of different water pools shoud be negligible. Lastly, Barbeta et al. (2020)
carried out a systematic experiment to study the isotopic offset between soil and stem water and found that differences are *"likely to be caused by water isotope heterogeneities within the soil pore and stem tissues…..than by fractionation under root water uptake"* (Barbeta et al., 2020).

Nevertheless – whether one agrees with the theory or not – the hypothesis had a significant impact in terms of i) questioning the comparability of ecohydrological studies because of methodological artifacts (e.g. mobile vs. bound soil water, soil and
plant water extraction methods, organic contamination), ii) testing existing and developing novel methods to investigate fundamental processes at the soil-vegetation-atmosphere continuum in an integrated manner; and finally, iii) questioning a number of concepts that have been applied since many years but now appear in a new light (e.g. root water uptake studies and the incorporation of isotope effects).

Consequently, many researchers have been focussing on these issues since and a number of publications have been pointing
out current limitations and ways forward (Berry et al., 2018; Bowling et al., 2017; Brantley et al., 2017; Dubbert et al., 2019b;

Penna et al., 2018; Sprenger et al., 2016). One of the most pressing issues identified is the establishment of a consistent, homogenized method for the analysis of water stable isotopes allowing for a solid analysis and interpretation of water isotopes in soils and plants and compare them with each other. Berry et al. (2018) postmarked current methods applied in ecohydrology as 'shotgun' methods, which is a suitable metaphor to descibe how many studies are carried out. What they call for is to

establish consistent and continuous methods of monitoring. Due to partially striking differences in $\delta$ values returned by different extraction methods (Millar et al., 2018; Orlowski et al., 2018b, 2018a), this is not an easy task. Even the (until recently) commonly accepted cryogenic vaccuum extraction (e.g. Koeniger et al., 2011) is being questioned frequently. On the other hand, novel methods based on isotopic equilibrium fractionation (e.g. Hendry et al., 2015; Wassenaar et al., 2008), outperform the extraction methods under certain conditions (Millar et al., 2018). As a consequence, a big question arises: Are

all source water studies biased?

### 1.1 Are ecohydrological source water studies biased? – The need for *in situ* methods

Certainly, not all source water studies are biased. Due to the systematic evaluations carried out in recent years and despite all the controversy on methodological aspects (Gaj et al., 2017; Millar et al., 2018; Orlowski et al., 2013; Orlowski et al., 2018; Orlowski, Breuer, & McDonnell, 2016; Orlowski, Pratt, & McDonnell, 2016; Thoma et al., 2018) it can be stated that in i)

soils that contain a high portion of sand (low portion of clay), ii) studies using isotopically labeled tracers ($^2H_2O$, $H_2^{18}O$), and iii) environments without water stress (low suction tension) the chance of methodological artefacts and the influence of additional isotope effects is at a minimum.

However, there are a number of isotope effects that clearly complicate the idealized situation where one takes a xylem sample from a tree (=unfractionated mixture of all water sources) in addition to sampling a soil profile (and perhaps groundwater) and

subsequently determines root water uptake depths. An updated view of the isotope effects potentially affecting water sources and consumers, depicted in Fig.1, emphasizes the sheer complexity that now is questioning many water-uptake studies.

**Figure 1**

In addition to the isotope effects summarized in Fig.1, there might be methodological alteration of the isotope composition caused by different extraction methods extracting different water pools and organic contamination causing an offset of $\delta$ values

when measured with laser spectroscopy (e.g. Barbeta et al., 2019; Martín-Gómez et al., 2015; Orlowski et al., 2016a).

The community seems to agree on three key challenges (Brantley et al., 2017; Dubbert et al., 2019b; Sprenger et al., 2016; Stumpp et al., 2018; Werner and Dubbert, 2016): i) develop consistent and comparable methods for a holistic monitoring of soil-plant-atmosphere interaction; ii) to further investigate, disentangle and quantify the abovementioned isotope effects by increasing the spatiotemporal resolution of water isotope measurements at the soil-vegetation-atmopshere interface; and iii) to

decrease the uncertainty when studying root water uptake by integrated measurements of sources and consumers into one framework. In other words, we need combined *in situ* systems for measuring both soil and xylem water isotopes in a higher spatiotemporal resolution in order to achieve an integrated analysis of soils and plants using the same methodology and

ultimately, measuring the same water pools (Sprenger et al., 2016). While it might be possible to achieve a high temporal resolution by destructive sampling, a number of disadvantages are associated with that: For instance, the experimental plot is

disturbed multiple times, small-scale heterogeneity might bias the outcomes and longer-term studies in a high temporal resolution are basically impossible. For plants, a high frequency of destructive sampling might harm the plant irreversibly. Lastly, when carrying out longer term studies the time and costs associated with destructive sampling and analysis might outbalance effort and benefits. Hence, *in situ* methods are essential for the detection, analysis and interpretation of related isotope effects (see Fig.1).

This review aims on summarizing recent advances in *in situ* water isotope measurement techniques for soils (depth-dependent and bulk soil) and plants (xylem and transpiration via physical leaf chambers). We begin with an overview of *in situ* studies in the compartments soils and vegetation. From thereon, we focus in separate chapters on main issues emerging from the existing sudies, namely i) materials and measurement systems, ii) calibration, standardization and validation and iii) comparability with water extraction studies and measurement of natural abundances of water isotopes. We then conclude and

propose ways forward in terms of a combined approach for a consistent, integrated method in order to study the temporal dynamics of processes at the soil-vegetation-atmosphere continuum.

## 2 Review: *In situ* approaches for measuring the soil and plant water stable isotope composition

### 2.1 *In situ* soil water isotope depth profiles

A number of early semi-*in situ* studies (pre-IRIS) exist, where researchers collected soil water vapor. For the sake of

completeness and acknowledging these pioneering efforts, those will be summarized briefly. Thoma et al. (1979) directed water from up to 25m depth through a molecular sieve, vacuum-trapped this water and determined the $\delta$ value for hydrogen ($\delta^2H$). The determined values agreed well with water extracted from a soil core. With a similar technique, Saxena and Dressie (1984) analyzed $\delta^{18}O$ from soil water vapor in profiles of up to 4m depth. Allison et al. (1987) sampled soil water vapor in glass jars. Though the shape of the isotope depth profiles for $\delta^{18}O$ and $\delta^2H$ was similar, the values did not match with those

obtained by cryogenic extraction. Izbicki et al. (2000) used a similar technique and achieved a better agreement compared to distilled core samples. It has also been shown that it is possible to sample soil gas from soil gas wells or probes and analyze the isotope composition of oxygen in $CO_2$ in the laboratory using classical IRMS methodology. This is possible due to the fact, that the molar abundance of oxygen in soil water is magnitudes higher than that in soil $CO_2$ and therefore $CO_2$ comes into equilibrium with soil water (see Stern et al., 1999).

With the introduction of IRIS, rapid progress in terms of continuous measurements and field deployable systems began. Koehler and Wassenaar (2011) were the first to show that unattended, continuous measurements of the water isotope composition of natural water samples (lakes, rivers, groundwater) based on isotopic equilibration between liquid and vapor phase are possible by using a gas permeable membrane contactor connected to a laser spectroscope. A similar gas permeable membrane system was tested by Munksgaard, Wurster, & Bird (2011). The first reported *in situ* measurement of soils was

reported by Herbstritt et al. (2012). A microporous hydrophobic membrane contactor was combined with an isotope laser spectrometer and tested for both pure liquid water and water that was directed through a soil column. The authors determined isotopic equilibrium fractionation factors for a range of temperatures by fitting the empirical factors *a, b and c* to the type-1 model of Majoube (1971, Eq.1):

$$\alpha = exp^{\frac{a\left(\frac{10^6}{T_k^2}\right)+b\left(\frac{10^3}{T_k}\right)+c}{1000}} \qquad (1)$$

where α is the isotopic equilibrium fractionation factor, $T_k$ is the temperature (in K), and *a, b*, and *c* are empirical parameters. Membrane-induced deviations from Majoube's (1971) prediction ranging from 0.27‰ to 0.64‰ for $\delta^{18}O$ and from 1.0‰ to 3.9‰ for $\delta^2H$ were reported. In addition, a vapor concentration correction similar to Schmidt et al. (2010) was conducted and recommended by Herbstritt et al. (2012). In the same year, Soderberg, Good, Wang, & Caylor (2012) presented the first 'real' (i.e. measured in the field) *in situ* data set from a semiarid environment. The authors aimed at investigating the validity of the Craig-and-Gordon (CG) model for soil evaporation and incorporating the effect of soil water potential on kinetic fractionation into the model, which they argue improves the model fit for very dry conditions. Their dataset was tested on a single profile of *in situ* measured values for $\delta^2H$ and $\delta^{18}O$ of soil water in a semiarid environment in Kenya. Soil air was drawn from several depths (5, 10, 20, and 30 cm) and directed to the laser spectrometer via buried Teflon tubing, with the final 10 cm of each tube perforated and packed with glass wool. Each depth was measured for 90 s. Soil temperature and soil water potential were measured at several depths. Though the *in situ* field data does not match with the data predicted by the CG model, the authors support the assumption of liquid–vapor isotopic equilibrium and propose a correction factor for dry soils in order to normalize 'free atmosphere' humidity to soil evaporation. Most likely, the non-matching $\delta$ values reported in the study were due to the dilution with a high amount of ambient air (400 ml/min) causing a contamination of the drawn soil air. A study on the effects of materials and methods for *in situ* water isotope measurements was presented by Pratt et al. (2016). While the first part deals with the optimization of the bag equilibration method (bag type, tubing, relative humidity) which is not scope of this review, the authors also compared *in situ* analyzed soil water vapor from depths of up to 180 m. The already existing HDPE tube at the two waste sites studied were screened with a 50 mm stainless steel mesh filter and the vapor drawn into the isotope analyzer (IRIS). The results of the *in situ* part of the study show partially large differences for depth profiles when comparing the field measurements with the ones obtained by the bag equilibration method (up to ±30‰ $\delta^2H$ and ±10‰ $\delta^{18}O$). In some depths, the agreement is better (±5‰ $\delta^2H$ and ±1‰ $\delta^{18}O$). Pratt et al. (2016) conclude that the *in situ* results of their study suggest that establishing natural, stable water isotope depth profiles for pore water in thick unsaturated mine waste is challenging.

Though insightful for testing the liquid–vapor isotopic equilibrium for continuous measurements and the effect of contactor membranes on isotopic equilibrium and fractionation factors, the approaches of the abovementioned studies were not applied further for soil water isotope measurements. Instead, two different types (or 'families') of gas permeable membrane probes evolved which both are based on similar principles but differ in design and level of complexity.

The first of these types of membrane systems was introduced by Volkmann and Weiler (2014) and thereafter used mainly be this research group for measuring soil and later also xylem water isotopes (compare chapter 2.3). The authors developed specific probes for the purpose of sampling soil water vapor. The main elements of these probes are a microporous membrane (Porex, Aachen, Germany), a mixing chamber and a sample, dilution and – optional – a throughflow line. The principle of operating the probes is based on drawing soil water vapor into the water isotope analyzer via the sampling line (30 – 35 ml/min). An automated system for non-destructive, high-resolution monitoring of soil water isotopes was proposed. This system can be operated in two modes, which the authors call advection-dilution sampling (ADS) and diffusion-dilution sampling (DDS). In ADS mode, air is simply drawn into the sample line, and dry air supplied at a lower rate via the dilution line, which causes i) soil water vapor to actively move into the tube (because of the slightly lower pressure inside the probe) and ii) lowering of the water vapor concentration of the whole system. In DDS mode, a throughflow line ending at the lower end (tip) of the probe is added to the system. This throughflow line allows to supply dry gas ($N_2$) to the system at a rate that is the difference of water drawn by the sampling line and supplied by the dilution line. Hence, the pressure difference between soil water vapor outside and inside the probe diminishes, and isotopic exchange occurs only via diffusion. Volkmann and Weiler (2014) were also the first to present validated natural isotope soil depth profiles (i.e. via bag equilibration; Hendry et al., 2015; Wassenaar et al., 2008). An acceptable agreement was achieved with their system. For the soil depth profiles, the authors report 95% limits of agreement of +1 ‰ (upper) and -1 ‰ (lower) for $\delta^{18}O$ and +6 ‰ (upper) and -6 ‰ (lower) for $\delta^{2}H$, both for ADS and DDS sampling methods compared to destructive bag equilibration isotope measurements. The range of measured $\delta$ values for $\delta^{18}O$ and $\delta^{2}H$ in soil water was further in the range of antecedent rainfall. Inspired by the system of Volkmann and Weiler (2014), Gaj et al. (2016) conducted an *in situ* study in a semi-arid environment, which can be seen as a proof of concept. In northern Namibia, the authors used commercially available polypropylene membranes (BGL-30, Umweltmesssysteme, Munich) and automated their system for measuring of soil water isotope-depth profiles up to 50 cm depth over multiple campaigns, different land covers (bare soil vs. vegetated) and different climatic conditions (dry and post-rain event) . Further, they are the first to study spatiotemporal differences in isotope depth-profiles with their *in situ* system.The profiles were compared to those obtained by cryogenic vaccuum extraction. While the shape of the isotope depth-profiles were in agreement and the precision of the *in situ* approach was good (0.8 and 2.5‰ for $\delta^{18}O$ and $\delta^{2}H$, respectively), there were partially large differences between the *in situ* data and the results based on cryogenic extraction: Between 15 and 50 cm, the RMSE was 3.9‰ for $\delta^{18}O$ and 9.2‰ for $\delta^{2}H$. For the shallow depths RMSE was as high as 7.0 ‰ for $\delta^{18}O$ and 43.4‰ for $\delta^{2}H$. Gaj et al. (2016) interpret these differences as potentially caused by an incomplete (cryogenic) soil water extraction, the time-lag between sampling soil water vapor and destructive sampling (day vs. night) , rayleigh fractionation caused by the uptake of air during the *in situ* measurement, natural processes (e.g. hydraulic redistribution), or natural heterogeneity. From today's perspective (author personal comment), it seems very likely that depleted atmospheric water vapor was drawn into the system at the upper soil depths causing the depletion of the *in situ* measurements. Further, condensation inside of the capillaries or contamination with organic substances could explain the observed differences. An important finding of this study is that for

the medium sand the authors studied, the standard deviation for $\delta^{18}O$ and $\delta^2H$ was inversely correlated to the soil water content,
i.e. the lower the water content of the soil, the higher the standard deviation and vice versa.

The second type of gas permeable probes originates from the study of Rothfuss et al.(2013) and has been applied in different forms and by different groups since then (see below). A major advantage of the gas-permeable membrane used (Accurel® PP V8/2HF, Membrana GmbH, Germany; 0.155 cm wall thickness, 0.55 cm i.d., 0.86 cm OD) is that the soil probes are cheap and can be built and customized easily by the user (e.g. length of exchange path; number, material and dimensions of the tubing/capillary inserted). The authors tested precision and accuracy of membrane-based *in situ* measurements in laboratory experiments. Rothfuss et al. (2013) set up an airtight acrylic vessel filled with fine sand, where a custom-made throughflow system with a gas-permeable polypropylene membrane was installed. Synthetic dry air was directed into the system, which during the passage isotopically equilibrates with the water of the surrounding sand. The authors used a mass flow controller to subsequently dilute the sample's water vapor concentration to 17.000 ppmv, which eliminates the dependency of measured $\delta$ values on water vapor concentration (Schmidt et al., 2010). They further investigated the effects of tubing material, soil temperature, sand water content and dry air flow rate as well as fast changes of source water $\delta$ values on measured vapor values ($\delta^2H_{vap}$ and $\delta^{18}O_{vap}$). Whereas measured sand $\delta^{18}O_{vap}$ was in good agreement, $\delta^2H_{vap}$ showed an enrichment relative to those determined at equilibrium according to Majoube, (1971) at all tested temperatures (8 – 24 °C). This was attributed to the used membrane and corrected for by fitting a linear regression in which the (known) liquid isotope standard value was estimated using temperature and the $\delta$ value of measured vapor (eq. 2a and 2b in Rothfuss et al., 2013). Rothfuss et al. (2015) proofed that the gas permeable membranes used are capable of delivering reliable isotope data over long time periods under laboratory conditions (in this experiment 290 days), though a proper validation of the measured $\delta$ values was not carried out (the authors compared the obtained $\delta$ values to those of the water intially provided to the soil column). Consequently, the same group presented several further studies employing these. Gangi et al. (2015) measured values of $\delta^{18}O$ in both soil water and carbon dioxide It was shown again that the membranes used (Accurel PP V8/2HF) did not lead to any isotopic fractionation and was suitable for combined measurements of $\delta^{18}O$ and $\delta^2H$. The experimental results were further modeled using MuSICA (Ogee et al., 2003). The authors proofed that it is possible to simultaneously study oxygen isotope exchange between soil water and $CO_2$ in natural soils which has an immense potential for constraining the atmospheric $CO_2$ budget. However, they state explicitly that further testing is required. Quade et al. (2018) conducted a study on the kinetic isotope fractionation of water during bare soil evaporation. The authors compared kinetic fractionation factors calculated with the widely known *'Keeling plot'* approach and an analysis of the *'evaporation line'* in dual-isotope plots applied on data obtained in a laboratory experiment. The results suggest limitations of the former approach, while the latter provided kinetic fractionation factors in the range of values reported in the literature (1.0132 ± 0.0013 for $\delta^2H$ and 1.0149 ± 0.0012 for $\delta^{18}O$). Quade et al. (2019) used the gas permeable membrane probes for partitioning of evapotranspiration of a sugar beet (*Beta vulgaris*) field. While soil water values (E) were measured *in situ*, the other required information for ET partitioning were obtained from Eddy Covariance measurements and destructive xylem samples (cryogenic vacuum extraction, 3-4h at 105°C measured via IRIS connected with micro-combustion module). Large discrepancies between the $\delta$ values of evaporation derived from destructive and non-

destructive measurements of soil water using a well-known transfer resistance model were found to cause significant differences in T/ET. Kühnhammer et al. (2020) monitored both soil and transpired water values for $\delta^{18}O$ and $\delta^2H$ *in situ* to

275 investigate root plasticity of *Centaurea jacea* (see chapter 2.2). Kübert et al.(2020) compared destructive vs. *in situ* methods for measuring the soil water isotope composition at a grassland site in southern Germany. Large mean absolute differences between cryogenic vacuum extraction and *in situ* measurements of 0.3–14.2 ‰ ($\delta^{18}O$) and 0.4–152.2 ‰ ($\delta^2H$) for soil liquid water were found with highest differences observed after irrigation with labeled water. Nevertheless, the authors see the *in situ* method as promising tool for future applications.

Another group from the United States developed a system for *in situ* measurements of soils and has applied the same type of gas permeable membrane probes in several studies (Oerter et al., 2017, 2019; Oerter and Bowen, 2017, 2019). In principle, the authors use the same methodology as presented by the group around Rothfuss, but provide a more flexible design of probes and a stand-alone solution for true field measurements (Figure 4). Their system – up to date – probably constitutes the most complete in terms of field deployability, calibration and the results reflect that (in particular see Oerter et al., 2017 and Figure

4 and chapter 3). The authors further present a novel approach for correcting their samples by including water and clay content (see chapter 3 calibration). In a primer, Oerter et al. (2017) used a vapor-permeable membrane technique and measured soil water isotopes *in situ* at four sites in North America and validated the water vapor probe method with the bag equilibration method, and vacuum extraction with subsequent liquid water analysis. The authors found that the accuracy of the three compared methods in their study is equivalent, with increased ease of use in its application, and sample throughput rates of 7

samples per hour by using the vapor probes. In fact, RMSE of the vapor probe method for $\delta^2H_{liq}$ values is lower than for the bag equilibration method in matching the $\delta^2H_{liq}$ values of vacuum-extracted soil water (1.7‰ for $\delta^2H_{liq}$ values and 0.62‰ for $\delta^{18}O_{liq}$). Hence, trueness for the vapor probe method in their application was greater than for the bag equilibration method. The analyzed profiles were used to investigate the effect of soil texture and the authors concluded that pedogenic soil horizons control the shape of the isotope profiles, which are reflective of local evaporation conditions in the soils.

**2.2 Soil and plant chambers for measuring the isotopic composition of evaporation and transpiration**

The isotopic composition of soil evaporation, transpiration and evapotranspiration can be measured *in situ* using laser spectrometers coupled to different chamber systems. These chamber-based *in situ* techniques were among the earliest development steps of *in situ* water isotope monitoring, well before the development of membrane-based approaches. There are two types of chamber systems to measure soil evaporation and plant transpiration fluxes and their isotope composition: flow-

300 through steady-state (Dubbert et al., 2013) and closed chamber systems (Wang et al., 2013). In a closed chamber the amount of water vapor will, upon closure of the chamber, increase over time, while the $\delta$ value of water vapor will change due to the continuous input of evaporated vapor. This method was first applied to measure the isotope composition of respired $CO_2$ (Keeling, 1958) and later adapted to determine the isotope composition of water vapor (evaporation and evapotranspiration, see Walker and Brunel, 1990). The Keeling (1958) plot approach is based on two assumptions. First, the isotope composition

of the source and background air are constant over the measuring period. Second, there shall be no loss of water vapor from

the system, e.g. due to dewfall. While this approach is generally not novel per se, prior to IRIS the water of the sampled vapor had to be collected in a cold trap and then measured using classical mass spectrometry. In an open chamber system, incoming (ambient background) and outgoing (mixed air inside the chamber) air are measured alternately and the isotope composition of evaporation (or transpiration or evapotranspiration) can be calculated by a mass-balance equation (von Caemmerer and Farquhar, 1981)

### 2.2.1 Soil chambers for measuring the isotope composition of evaporation

The isotope composition of soil evaporation have been predominantly used to achieve a better understanding of the dynamics of hydrological processes (Braud et al., 2005b, 2005a, 2009b, 2009a; Haverd et al., 2011) and to partition evapotranspiration into its components: soil evaporation and plant transpiration (e.g. Dubbert et al., 2014b, 2013; Haverd et al., 2011; Rothfuss et al., 2012, 2010; Williams et al., 2004; Yepez et al., 2007, 2003). Prior to the development of IRIS there were few studies measuring $\delta^{18}O_E$ directly using cold trapping methods under controlled conditions and - to the best of our knowledge - no observations under natural conditions. Instead, researchers relied on the CG model (Craig and Gordon, 1965), predicting the isotope composition of evaporated vapor based the source (=soil) water (e.g. Brunel et al., 1997; Wang and Yakir, 2000). Therefore, first approaches to combine soil gas-exchange chambers and laser spectrometers, concentrated on conducting sensitivity analysis of the CG model towards its input parameter (relative humidity, soil temperature, soil water content and the isotope composition of soil water and atmospheric vapor (Braud et al., 2005b, 2005a, 2009b, 2009a; Dubbert et al., 2013; Haverd et al., 2011; Rothfuss et al., 2010, 2012). Conclusively, the correct estimation of the evaporation front is particularly important. Usually the soil layer with the highest $\delta$ value is associated with the location of the evaporation front but therefore the vertical discretization of the measured soil should be as small as possible (2 cm or less in the upper soil). Sensitivity studies revealed that precise parameterization of the environmental conditions at the evaporation front, which may diverge tremendously within a few cm of soil depth (up to 8 ‰ from the soil surface to 5 cm depth), is pivotal for correct predictions of $\delta^{18}O_E$ (Dubbert et al., 2013). This is true particularly in arid regions, where dry periods without any precipitation can last several months and the evaporation front can be located in deeper soil layers (Dubbert et al., 2013; Gaj et al., 2016). This does not only highlight the value of direct *in situ* estimates of $\delta^{18}O_E$, but also the deployment of *in situ* soil water isotope measurement set ups when using the CG model. Only spatially high resolved continuous *in situ* soil water observations meet the desired requirements necessary to resolve the evaporation front.

In any case, direct *in situ* measurements of soil evaporation are mostly limited to laboratory studies conducting sensitivity analysis of the Craig and Gordon model and its input parameters as well as calculating kinetic fractionation (for a recent paper see Quade et al., 2018). It is often not technically possible to observe $\delta^{18}O_E$ of undisturbed vegetated soil in the field. However, the isotope composition of evaporated vapor from bare soil patches differs significantly from that of evaporation from soil with vegetation cover (particularly in grasslands, see Dubbert et al., 2013). Therefore, even at present $\delta_E$ is mostly modeled using the CG equation in ecosystem studies.

## 2.2.2 Plant chambers for measuring the water isotope composition of Transpiration

Direct estimation of $\delta$ values of plant transpiration ($\delta_T$) has been difficult prior to the development of IRIS. Nevertheless, some studies used cold trapping methods and gas exchange systems to estimate $\delta_T$ (e.g. Harwood et al., 1998) and ecosystem evapotranspiration (e.g. Yepez et al., 2007, 2003) via keeling plots (Keeling, 1958). The main limitation of these early studies – similar to those for soil water isotopes – was the spatiotemporal resolution. With the advent of laser technology, studies multiplied coupling laser spectrometer to gas-exchange systems of different scales (i.e leaf to canopy level) to assess the isotope composition of ecosystem water fluxes (see Wang et al. (2012) for the first *in situ* observation of $\delta_T$). Nowadays, open chamber systems are predominantly used to measure the the isotope composition of transpiration *in situ* (see chapter soil chambers).

*In situ* observations of $\delta_T$ had an immense impact on ecosystem partitioning studies, as they have the advantage of directly measuring the transpiration signature, while destructive sampling techniques observe xylem or leaf water values, essentially involving a modeling step to obtain $\delta_T$. A number of ecosystem partitioning studies (e.g., Griffis et al., 2010; Williams et al., 2004; Yepez et al., 2003) even simplified by assuming isotopic steady-state (the isotope composition of xylem=transpiration), although there is growing evidence that plants rarely reach isotopic steady-state throughout the day (Dubbert et al., 2014a, 2017; Simonin et al., 2013). Therefore, assuming isotopic steady-state for the purpose of evapotranspiration partitioning will largely depend on the desired temporal scale (considering non steady-state definitely necessary at sub-diurnal to diurnal scale but unimportant at larger time scales, i.e. weeks or months). In case non steady-state is likely to occur, $\delta^{18}O$ of transpiration can be modeled using a Dongmann style version of the CG equation (Dongmann et al., 1974). However, this complicates the partitioning approach tremendously in comparison to direct chamber measurements of the transpiration water isotope composition, as a large number of additional observations are necessary (in particular, stomatal conductance and Transpiration rates). Another important consideration in regard to the method of choice (*in situ* transpiration measurements vs. modeling) is the possibility to sample unfractioned xylem water. For example, herbaceous and grass or agricultural species do not have suberized stems and destructive sampling would rely on leaf water sampling or sampling the plant culm belowground, which is highly destructive and not possible on normal plot sizes. Moreover, while the majority of studies still provide evidence for an unfractioned uptake and transport of xylem water through plants, there is growing evidence of fractionation of xylem water during times of limited transpiration rate (drought conditions, for deciduous species, see e.g. Martin-Gomez et al. 2017). Similar to *in situ* soil evaporation isotope observations, *in situ* observations of $\delta_T$ have further been used to advance our understanding on water isotope fractionation at the leaf level (e.g. Dubbert et al., 2017; Piayda et al., 2017; Simonin et al., 2013; Song et al., 2015a, 2015b, 2013). For example, we have seen that the leaf water turn-over time which can effectively be described by stomatal conductance and leaf water volume, is extremely species-specific spanning from several minutes to several hours (Song et al., 2015a). As the leaf water turn-over time describes the necessary time for a leaf to reach isotopic steady-state after a change of ambient conditions (see Simonin et al., 2013; Song et al., 2015a), isotopic steady-state can either

be observed for large parts of the day (e.g. in many herbaceous species) or not at all (e.g. in plant species strongly controlling their stomatal conductance, see Dubbert et al., 2017, 2014a for an overview).

Direct *in situ* estimates the isotope composition of transpiration have also been used to derive root water uptake proportions (Kühnhammer et al., 2020; Volkmann et al., 2016b) by assuming isotopic steady state and substituting $\delta^{18}O_X$ (xylem) with $\delta^{18}O_T$. Recently, Kühnhammer et al. (2020) used a classical isotope mass balance approach (SIAR) and added physiological

restraints by combining soil and plant water potentials to derive more physiologically accurate root water uptake proportions and plant reactions to water availability changes in different depths. However, given the often very likely violation of steady state assumptions under natural field conditions, this can be recommended only under very tightly controlled laboratory conditions and knowing the leaf water turn-over time.

### 2.3 *In situ* measurements of plant xylem water isotopes

For the direct measurement of plant xylem water isotopes, only two studies are reported up to date. Volkmann et al. (2016) present field observations of the xylem water isotope composition of two adult field maple trees (*Acer campestre L.*) obtained over several days during a labeled irrigation event using IRIS. The obtained *in situ* data was compared against results from destructive sampling with cryogenic extraction. Similar to their *in situ* soil measurements, Volkmann et al. (2016a) used the same membrane system to infer the isotope composition of xylem water. Several holes were drilled into each of the target trees

and the gas permeable membranes inserted into those. In order to prevent the intrusion of atmospheric air the outside was sealed with silicone. Similar to the soil studies, dry gas (here $N_2$) is provided by a throughflow line and directed to the laser spectrometer via the suction of its vacuum pump.

### Figure 2

With the obtained data Volkmann et al. (2016a) demonstrated that temporal changes as well as spatial patterns of integration

in xylem water isotope composition can be resolved through *in situ* measurement. In both studied trees, diurnal cycles of xylem water isotopes were found. However, the authors could not prove whether this is a true diurnal cycle or introduced through imperfect accounting for temperature-dependent liquid–vapor fractionation at the probe interface. The authors achieved a median precision of 1.1‰ for $\delta^2H$ and 0.29‰ for $\delta^{18}O$ values (1σ) for an integration period of 120 s. When comparing the *in situ* measured xylem $\delta$ values (IRIS) with the results obtained from destructive sampling (measured with IRMS) a significant

correlation was found for both water ($\delta^2H_{IRIS}=1.26\times\delta^2H_{IRMS}+14.51$, $r^2=0.86$, P<0.0001, $\delta^{18}O_{IRIS}=0.91\times\delta^{18}O_{IRMS}-4.87$, $r^2=0.46$, P<0.001, robust BSquare-weighted M-regression). However, when taking a closer look at the agreement of *in situ* and destructive data, partially high uncertainties are apparent (see Fig.2, reprinted with permission). In addition, the uncertainty (especially of the *in situ* data) is up to 20 ‰ for $\delta^2H$ and up to 3 ‰ for $\delta^{18}O$.

Marshall et al. (2020) tested an alternative method for measuring the isotope composition of tree xylem and showed that both

natural abundances and highly enriched $\delta$ values (labeling experiment) can be monitored *in situ* over more than two months. Their approach is based on drilling a hole (which the authors refer to as stem borehole) laterally through the complete trunk of

a tree and connecting both ends with tight fittings to the manifold system and the laser spectrometer. The temperatures within the boreholes were monitored using thermocouples and later used for vapor-liquid conversion of the measured $\delta_{vap}$. The authors tested their system on two occasions on pine trees: i) in a cut-stem experiment and ii) in a whole-root experiment. They further developed a model to test the feasibility and limits of the borehole method. This included the estimation of the time constants for diffusion of water vapor to and from the borehole wall, and for the passage of the flowing airstream and the centre of the borehole (i.e. isotope exchange during the passage of air through the borehole) as well as the prediction of the water isotope composition. For both experiments, Marshall et al. (2020) found close agreement of the source water provided to the trees, the ones measured in the stem boreholes and the ones predicted by the model. In the cut-stem experiment, it took several hours after a change in water source before this agreement was reached. In the case of the intact-root experiment, it took almost two weeks until source and measured $\delta$ values agreed. In this experiment, the authors further tested equipping the tree with two stem boreholes. For the bottom borehole, the deviations to source water values were nearly zero, meaning that the derived $\delta$ values of xylem water agreed with the source water values for both natural abundance ($\delta^{18}O$ = -0.1 ± 0.6 (SD) ‰, $\delta^{2}H$ = 1.8 ± 2.3 ‰) and the label phase ($\delta^{18}O$ = -0.25 ± 0.22 ‰, $\delta^{2}H$ = 0.09 ± 7.8 ‰). In contrast, the top borehole showed systematic deviations from source water values for both $\delta^{18}O$ and $\delta^{2}H$. $\delta^{18}O$ xylem values were depleted in $^{18}O$ in relation to source water by -2.8 ± 1.5 ‰ and -3.9 ± 0.3 ‰ for the natural abundance and label phase, respectively. In contrast, $\delta^{2}H$ xylem values were enriched in $^{2}H$ as compared to source water by 5.3 ± 3.0 ‰ and 1.9 ± 8.5 ‰. Figure 3 shows the results from their intact-root experiment.

**Figure 3**

With the additional measures taken and the developed model, Marshall et al. (2020) suggest that this deviation was due to non-equilibrium conditions in the upper borehole due to its small diameter (relative humidity of sample air was 98 ± 2 % for the bottom borehole and 88 ± 3 % for the top borehole). Finally, the authors measured the sap flow velocity, which was 0.97 ± 0.4 cm/hr based on the sap probes (heat-ratio method). The time lag between both boreholes yields 1.08 cm/hr on average, which is in agreement to the sap flow estimates.

Concluding this chapter, Table 1 provides information on all reviewed studies, details on the setup, main findings as well as advantages and disadvantages of the applied methodologies.

**Table 1**

## 3 Setup, Calibration and Validation of *in situ* measurements of the soil and plant water isotope composition

Apparent from the review of studies is that *in situ* measurements are still in development stage; hence, applied methods and approaches vary greatly. In this chapter, we pick out key aspects that need to be considered and propose a way towards more comparable and homogeneous setups. The biggest and most critical issues emerging from the existing studies are i) the materials and approaches used for sampling the water vapor; ii) the calibration of the system iii) the avoidance of condensation and iv) how to validate the *in situ* data compared to other methods and how to interpret it best. We focus in this chapter on methods for obtaining *in situ* depth profiles of soil water isotopes and the measurement of xylem water isotopes due to the fact that methods for monitoring bulk soil evaporation and transpiration at the leaf level have been discussed previously in detail (Soderberg et al., 2012, Song et al., 2015).

### 3.1 Materials and approaches for sampling soil water vapor

Most of the reviewed studies used gas permeable membranes (e.g. Accurel PP V8/2HF, Membrana GmbH; 0.2-µm porosity, 0.155 cm wall thickness, 0.55 cm inner diameter, 0.86 cm outer diameter) with an inlet and outlet (e.g. Gaj et al., 2016; Oerter et al., 2017; Rothfuss et al., 2013; Volkmann and Weiler, 2014). Some groups built the probes themselves (Oerter et al., 2017; Rothfuss et al., 2013), some used more complex custom-made parts (Volkmann and Weiler, 2014), others used factory-made probes (Gaj et al., 2016). The important point is that with all of these membrane systems it was shown that no isotopic fractionation occurs due to the membranes; hence this type of probes is suitable in general. Self-made soil gas probes are much cheaper and can be adjusted to the application (i.e. length of exchange path, number and position of in- and outlets, size and material of capillary/tubing connected). To direct sampled water vapor to the analyzer, tubing materials used should ideally be hydrophobic, gas-tight and isotopically inert.

The number of in- and outlets of the probes depends on the measurement approach. In general, two of these exist: i) a pull-only system (e.g. Volkman and Weiler, 2014), where water is drawn simply through the gas-permeable membrane by the force of the vaccum pump of the laser spectrometer. Such a system in fact requires only one capillary and thus is the simplest of the setups. However, it should be considered that a notable amount of air is drawn from the media to be measured (soil/plant). This could be especially relevant for applications in tree xylem as it might increase the risk of cavitation and hence, damage the plant. A pull-only method might not even be possible in trees due to the different structure of xylem compared to soil. The extracted volume of soil water vapor can easily be calculated by multiplying the flow rate with the measurement time. Most studies use ii) probes with two capillaries: one in- and one outlet (e.g. Oerter et al., 2017; Rothfuss et al., 2013). This changes the approach drastically, because now dry air is pushed through the inlet (via a dry gas supply) entering the membrane from one side and leaving it at the outlet. During the passage of the dry air, water from the soil air diffuses into the membrane and exchanges isotopically through the gas-permeable membrane. Unless soils are extremely dry, saturated sample air can be assumed to be in isotopic equilibrium with liquid soil water. However, the isotopic equilibrium fractionation factor could be affected by soil water tension (Gaj and McDonnell, 2019) as well as wettability, texture and chemical composition of soil

surfaces (Gaj et al., 2019). Directing air into the system in the push-through method has two consequences First, one needs to get rid of this excess air before it enters the laser spectrometer to avoid damage. This is commonly achieved by an open-split just before the analyzer inlet. Second, the chance of external water vapor entering the stream can be excluded, as long as air is coming out of the open-split, which is a clear advantage over the push-only method, where it needs to be assured that all connections are air-tight.

The pull-only system can also be operated with an additional inlet capillary/tube connected to a reservoir with drying agent. Doing so, atmospheric or dry air (via passage through a drying agent) is drawn into the gas permeable probe and equilibrated therein during the passage. Flow rates, however, are not adjustable using this approach.

It needs to be carefully decided which approach to use and, ultimately, this depends on the application (e.g. tracer test, measuring natural abundances, long- vs. short-term measurements). A pull-only system is technically much easier to build, install and maintain and also cheaper, but it it is critical to avoid external air to enter the system at any of the connections. The push-through approach is more flexible and flow rates can be adjusted, but it requires more maintenance, connections (for provision and control of dry air at the inlet), and valves.

Figure 4 depicts a schematic of an *in situ* soil water isotope system (reprinted with permission from Oerter and Bowen 2017).

**Figure 4**

**3.2 Saturation of water vapor, condensation and dilution**

Condensation (or better: avoidance of it) is the most critical practical issue for all *in situ* approaches, regardless if soils or plants are measured. If condensation occurs inside of the tubing or inside the chamber, the $\delta$ values measured will be subjected to Raleigh fractionation and hence, do not represent the isotope composition of the medium that is to be measured. Hence, it needs to be assured that the water vapor pressure in the sampling line never exceeds the saturation water vapor pressure or that condensed water is removed from the system. In the reviewed studies, condensation during measurements is dealt with in two different ways:

i) Dilution with dry air directly in the membrane system (Volkmann et al., 2016a; Volkmann and Weiler, 2014) or shortly after (Oerter et al., 2017; Oerter and Bowen, 2017, 2019; Rothfuss et al., 2013, 2015, Kühnhammer et al. 2020). This way, the water vapor concentration of the system is lowered and condensation less likely.

ii) Heating of the tubing (suggested by Gaj et al., 2016). Assuring that the temperature of the transport line is always warmer than the temperature at the sampling location will avoid condensation to occur. Even in warm climates this might be necessary as solely the temperature difference between the location where water vapor is equilibrated (i.e. inside of the gas permeable probe) and the sampling line is decisive if condensation occurs or not (refer to section recommendations for further elaboration on this issue).

Flushing the system with dry air prior to the measurement removes water that condensated before the current measurement (Kühnhammer et al., 2020; Volkmann and Weiler, 2014). An ideal system would include different measures to automatically

ensure the prevention of condensation both during and in between measurements. During measurements condensation is
prevented by dilution with dry air and heating of tubing prior to that point. A three-way valve directly after the measurement
point could be included to remove liquid water from the gas permeable tubing/borehole without having to pass it through the
whole system. In between measurements it could be used to cut off the measurement point from the rest of the system while
decreasing the relative humidity from the sampling point to the analyzer via the dilution line.

Condensation occurs whenever the temperature inside the sampling line (e.g. inside of a soil gas probe) is cooler than on the
outside (e.g. atmospheric air). This is often likely and will affect the water isotope data tremendously. Hence, it is of utmost
importance to include measures to avoid it into the sampling design, check measurements for it regularly and best to avoid it
altogether. However, it is not always easy to identify. For this reason, we present three examples of (raw) isotope measurements
in Fig. 5 which depict i) a 'good' measurement cycle; ii) a measurement cycle initially influenced by condensation, but then
turning into a clean measurement once the condensation disappears and iii)/iv) bad measurement cycle with condensation
affecting the complete data. Fig. 5 shows extracts from data collected by the authors during a field campaign in Costa Rica in
the beginning of 2019.

**Figure 5**

**3.3 Calibration protocols**

The calibration of water $\delta$ values is a crucial point, and it is more complex and error-prone when measuring water vapor
isotopes *in situ* compared to liquid water samples. It is generally comprised of the following steps: i) Standard preparation; ii)
Correction for water vapor concentration dependency of the raw $\delta$ values; iii) Specific corrections (mineral mediated
fractionation, organic contamination, carrier gas and biogenic matrix effects); iv) Drift correction; v) Conversion from vapor
to liquid values; and finally, vi) Normalization to VSMOW scale. There is a great variety on how (and even if) each of these
steps were addressed in the reviewed papers. The subsequent section summarizes the key points in terms of calibration
procedures. We then put a special focus on the approaches presented by Oerter et al. (2017) and Oerter and Bowen (2017),
who propose a novel, innovative method for the calibration of *in situ* measurements of soil water isotopes.

**3.3.1 Soil water vapor isotope standards**

Ideally, isotope standards are prepared in the same medium that is measured. That means, one should use soil standards when
measuring soil water isotopes and use water standards when measuring liquid water samples, as well as use the same probes
(e.g. membrane material) and sample flow rates. Gaj and McDonnell (2019) provided empirical evidence that soil matrix
effects can affect the fractionation factors in soils and need to be accounted for. The clear advantage of this is that such mineral-
mediated isotope effects can be incorporated into the calibration procedure using soil standards in a way that the standard will
be affected in the same way as the measured sample. However, one might also argue against this as pre-drying the soil (e.g. at

105°C) might destroy the soil matrix. Further – and again most pronounced in clay-rich soils – such a pre-drying might not remove all water (Gaj et al., 2017) and hence, create an isotopic offset into the soil water standards. Another disadvantage is that the preparation of soil standards requires more practical effort. Soil from the site of interest needs to be collected, oven-dried and placed in suitable standard bags or containers. Subsequently this soil needs to be spiked with the isotope standards (Gaj et al., 2016; Oerter et al., 2017; Oerter and Bowen, 2017; Rothfuss et al., 2015). Ideally, soil from different horizons is used for that as well, because the soil texture and, hence, isotope effects might change throughout the soil profile (Oerter et al., 2017; Oerter and Bowen, 2017). In addition, a range of water contents should be covered in the calibration process. This makes the calibration using soil standards labor-intensive and multiplies the number of standards to be measured (different soil horizons x different standards x different water contents). Soil structure might also affect the measured $\delta$ values (Oerter et al., 2014). However, due to the necessity of destructive sampling and drying for standard preparation, this effect can hardly be accounted for. In contrast, using water standards for calibration is rather straightforward, as only different water vapor concentrations need to be considered for calibration. This can be done either using a system for vapor injection (e.g. a standards delivery module or nebulizer) or simply placing the water standards in bags or containers and measuring the headspace. In the latter case, calibration of water vapor concentrations needs to be controlled via diluting the sampled water vapor with dry air to obtain lower water vapor concentration values. The big disadvantage of using water standards is that soil induced isotope effects are not incorporated at all and this can lead to notable errors in the corrected $\delta$ values later on. Hence, for best isotope data we recommend soil standards when measuring soil water isotopes (depth profiles and evaporation) and water standards when measuring *in situ* plant water isotopes (transpiration and xylem) or atmospheric water vapor.

In regard to chamber based measurements, correction has mostly been done with liquid standards injected into the instrument in the past. However, when integrating chambers in a larger *in situ* framework, we recommend to use water equilibration standards instead. Obviously, the background dry-gas is of major importance here, as the air matrix of the standard should be the same as that of the sample.

### 3.3.2 Correction for water vapor concentration

Because of the influence of different water vapor concentrations on measured $\delta$ values (Lis et al., 2008, Picarro, 2015; Schmidt et al., 2010), a correction needs to be performed. A linear best-fit equation can be derived if a standard of known $\delta$ value is measured at different water vapor concentrations. The slope and intercept of the best-fit line through these points are the two values that are used to post-process vapor delta values with variable water concentration (Picarro, 2015).

Schmidt et al. (2010) investigated concentration effects on IRIS $\delta^{18}O$ and $\delta^{2}H$ measurements in detail and showed a positive correlation of the water vapor concentration with $\delta$ values. In their study, the authors report a concentration effect of 1.2 to 1.4‰ per 10.000 ppmv for $\delta^{18}O$ and 0.6 ‰ per 10.000 ppmv for $\delta^{2}H$. The precision of the IRIS instrument used did not change over the range covered (5000 ppmv to 30000 ppmv). They proposed to measure the isotope composition at the same water vapor concentration or to correct raw values for water vapor concentration dependency (before applying any other correction).

The instrument-specific connection of raw $\delta$ values with the water vapor concentration of the measured sample should be investigated by e.g. measuring different water vapor standards at different dilution rates.

The water vapor concentration when carrying out *in situ* measurements of the isotope composition of soil and xylem water is affected by the temperature of the media of interest but also soil moisture or stem water content. Indirectly, the flow rate chosen by the user also affects the water vapor concentration (if flow rates are too high, saturation will not be reached). The interplay of those factors is complex and not trivial to account for (refer to chapters 3.4 and 5 for elaborations on this issue). In soil and leaf chambers, relative humidity and vapor pressure deficit affect the water vapor concentration of the measureand.

### 3.3.3 Other corrections (mineral mediated fractionation, organic contamination, carrier gas and biogenic matrix effects)

Recent research has shown that especially in clay-rich soils, an offset in comparison to water used for spiking can be observed due to tightly bound water (Gaj et al., 2017; Newberry et al., 2017; Oerter et al., 2014). This creates a real challenge for any soil water isotope measurement and was discussed heavily (Orlowski et al., 2013, 2016b, 2016a; Sprenger et al., 2016). It has to be noted, that those studies investigated destructively sampled and therefore unstructured soils. Under natural conditions soil structure might however play a significant role in soil-intern isotopic differences. Up to date, it is not clear how to best handle these additional factors. As stated above, a preparation of isotope standards in the same soil that is to be measured seems to be the most promising approach, and Oerter et al. (2017) provide an innovative procedure to calibrate their data (see chapter 3.4).

In addition, spectral contamination of IRIS measurements caused by organic compounds has been discussed frequently and was recognized as a major source of error when extracting water from plant tissues (Barbeta et al., 2019; Brand et al., 2009; Brantley et al., 2017; Hendry et al., 2011; Martín-Gómez et al., 2015; Millar et al., 2018; Newberry et al., 2017a; Penna et al., 2018; West et al., 2010, 2011). It is not known up to date, if this plays a role for *in situ* approaches (refer chapter 5). Volkmann et al. (2016b) speculated in their study that organic contamination might be one of the reasons for the observed discrepancies in their dataset. For liquid water samples, a method for correcting for the influence of organic substances exists (Barbeta et al., 2020; Lin et al., 2019; Schultz et al., 2011; Wu et al., 2013). Thereby, deionized water is spiked with varying amounts of methanol and ethanol to create correction curves for $\delta^{18}O$ and $\delta^2H$. An adaptation of this method is theoretically feasible for water vapor measurements, but has not been tested thoroughly until today (personal communication, M. Hofmann, Picarro). It should be noted, however, that methanol and ethanol are not the only possible contaminants and others might additionally influence the absorption spectra. Generally, it is advisable to perform a check if organic contamination for the particular set of samples is an issue using the pertinent software (e.g. Chemcorrect). If this is the case, measuring plant samples and samples from the upper soil layers with mass spectrometric analysis or corrections are required.

Finally, the issues of carrier gases and biogenic matrix effects have been raised recently. Gralher et al. (2016) tested how different mixtures of $N_2$, $O_2$ and $CO_2$ as carrier gas affected water stable isotope composition. With increasing $CO_2$ and $O_2$ concentrations, they report linearly increasing and decreasing values for $\delta^{18}O$ and $\delta^2H$, respectively. As those concentrations

would have to be determined separately, the authors used the line width related variable, one of the instruments spectral variables, as a representative term of the gas composition and provide an equation for a straightforward correction of $\delta$ values. Gralher et al. (2018) tested the effect of inflation atmosphere (dry air vs. $N_2$) and accumulation of biogenic gas ($CO_2$ and $CH_4$) with longer storage times on the bag equilibration method to measure pore water stable isotopes. They found that microbial production of $CO_2$ increasingly impacted the water isotope composition with longer storage and conclude that instrument-

specific post-correction yielded more reliable results when using dry air instead of $N_2$.

### 3.3.4 Drift correction

As for the measurement of liquid water samples, it is recommended to always use a drift standard that can be measured either after each run (e.g. after measuring one soil profile or a set of tree replicates) or after a certain time. A linear correction similar to the regression for water concentration can than be performed.

### 3.3.5 Conversion of vapor to liquid water isotope composition

All of the presented studies are based on isotopic exchange between the air outside and inside of the gas permeable probe. Ideally, equilibrium fractionation is achieved during the passage of the air through the membrane. The isotope composition of water (soil or xylem) can then be calculated applying the well-established equations for equilibrium fractionation (see Horita and Wesolowski, 1994; Majoube, 1971):

$$\propto {}^2H = \exp^{\frac{\left(a*\left(\frac{10^6}{T^2}\right)+b*\left(\frac{10^3}{T^1}\right)+c\right)}{1000}} \tag{2}$$

$$\propto {}^{18}O = \exp^{\frac{\left(a*\left(\frac{10^6}{T^2}\right)+b*\left(\frac{10^3}{T^1}\right)+c\right)}{1000}} \tag{3}$$

$$\delta {}^2H_{liq} = \propto * (1000 + \delta {}^2H_{vap}) - 1000 \tag{4}$$

$$\delta {}^{18}O_{liq} = \propto * (1000 + \delta {}^{18}O_{vap}) - 1000 \tag{5}$$

where $\alpha$ is the fractionation factor, T is the temperature in Kelvin and $\delta_{vap}$ and $\delta_{liq}$ the value of water vapor and liquid water,

respectively. The empirical factors a, b and c are tabulated in the above cited literature and commonly used as a= 28.844, b= -76.248, c=52.612 for $\delta^2H$ and a=1.137, b=-0.4156, c=2.0667 for $\delta^{18}O$ (Majoube, 1971). As per equations 2 and 3, the temperature is needed for this conversion. Hence, it needs to be measured at the location of exchange (e.g. at the gas permeable probe). A conversion of vapor to liquid values is also possible when the water vapor is not saturated and in isotopic equilibrium (via equal treatment principle of isotope standards), but is not recommended because, for soils, for example, the isotope

standards would be needed to be prepared with the exact soil moisture and temperature as the sample to be measured. This becomes very laborious because soil water contents are highly variable with depth and time. The *in situ* soil water isotope setup of Rothfuss et al. (2013) showed deviations of $\delta^2H$ in the vapor phase as compared to expected equilibrium fractionation using the equations defined in Majoube (1971). They argue that this difference arises from either the purging (we are not

measuring in a closed system) or an isotopic effect of the membrane material and propose specific equations for converting
vapor to liquid phase $\delta$ values for this type of setup.

The final step is – similar to liquid water isotope measurements – the normalization to VSMOW scale: (we spare the procedure here as this is widely known and sufficiently documented).

## 3.4 Validation – comparing apples and pears?

As shown in the previous chapter, calibration protocols for addressing the abovementioned steps vary greatly. Not always all the steps are addressed – either because it was not relevant for the particular investigation or because it was simply neglected. Thus, it is necessary to introduce a way of assessing the measurements. Across studies, trueness, precision and reproducibility of *in situ* methods are generally good. For an evaluation of accuracy, the reviewed publications compared the obtained isotope composition either with cryogenically extracted samples (Gaj et al., 2016; Soderberg et al., 2012; Volkmann et al., 2016a; Volkmann and Weiler, 2014), results from direct bag equilibration methods (Pratt et al., 2016) or both (Oerter et al., 2017). Further, theoretical approaches (mass balance calculations and modeling) have been applied to reproduce the *in situ* measurements (Rothfuss et al., 2013, 2015; Soderberg et al., 2012). The agreement of soil profiles extracted with vacuum extraction at deeper soil layers is generally better. In the upper soil layers, partially large differences (> 10 ‰ in $\delta^2$H) are encountered. Possible reasons include contamination with organic compounds or interference with atmospheric air when using a pull-only system as well as mineral-mediated effects. In the light of recent findings suggesting that water from cryogenic vacuum extraction and *in situ* approaches represent different water pools (Orlowski et al., 2016b; Sprenger et al., 2016), this way of validation might not be suitable. Instead, validating *in situ* data with the established bag equilibration method by deploying aluminium or other air-tight bags and measuring the headspace air, should deliver true means of comparing the data (see Oerter and Bowen, 2017).

The validation of the xylem water isotope *in situ* measurements of Volkmann et al. (2016a) yielded good results in terms of precision (median of 1.1‰ for $\delta^2$H and 0.29‰ for $\delta^{18}$O) and reproducibility (median of 2.8‰ for $\delta^2$H and 0.33‰ for $\delta^{18}$O). Diurnal variations in both isotopes did not correlate with those of temperature estimates for the different probes; hence, the authors recommend measuring the temperature inside of the probe in the future. They further state that when comparing the values obtained *in situ* with cryogenic extractions and subsequent measurement using IRMS, a significant correlation between the two exists. For data collected before the application of labeled irrigation, they achieved a good agreement and little systematic difference for $\delta^2$H (0.9 ±1.8‰). For $\delta^{18}$O, a clear inter-method bias of -4.3± 0.7‰ was found. The discrepancy in their data was hypothetically attributed to contamination by volatile organic compounds (VOC's), lateral mixing (through intervessel pits), axial dispersion and the time lag between irrigation water arrival at the twig/crown versus trunk level.

The closest agreement of the reviewed manuscripts when comparing *in situ* derived data with other methods was achieved in the study of Oerter et al. (2017). Both in terms of measurement and data handling, their methodology appears to be the most complete at present. In addition, the authors propose a novel, innovative way of calibrating *in situ* data of soil water isotopes. Oerter and Bowen (2019) proposed an updated approach including the correction for carrier gas effects and also introducing

the installation of soil water isotope probes in direct contact with roots/the rhizosphere. A reprint of their isotope depth-profiles determined with gas-permeable soil gas probes, direct equilibration and vacuum-extracted profiles are shown in Figure 6.

**Figure 6**

We propose here an adaptation (more general) of the procedure used by the authors:

i) collect samples from each soil depth interval from the site of interest and dry soil in oven, place samples in gastight bags or containers (e.g. 0-10 cm, 10-50 cm, > 50cm);

ii) add different amounts of isotope standard with known $\delta^2H$ and $\delta^{18}O$ values to obtain a range of water contents (e.g.
5%, 10%, 20% water content x 2 standard solutions x 3 depth intervals = 18 calibration bags or, ideally, undisturbed soil core samples from the site of interest);

iii) add soil temperature sensors to standard bags/containers;

iv) measure standard preparations under a range of temperatures (e.g. 0 -35°C);

v) perform multi-linear regression analysis (e.g. nlme package in R) in order to estimate theoretical liquid water
standard values using the parameters *measured vapor value ($\delta^{18}O_{vap}$ and $\delta^2H_{vap}$), soil moisture content (GWC) and temperature (TEMP);* other parameters such as clay content or water vapor concentration might be added ;

vi) selection of best fit equation for estimation of $\delta^2H_{liq}$ and $\delta^{18}O_{liq}$ of the isotope standards (in Oerter et al., 2017: $\delta^{18}O_{liq} = 9.954 - 0.163 \times TEMP + 0.002 \times TEMP^2 + 13.386 \times GWC + 1.051 \times \delta^{18}O_{vap}$; $\delta^2H_{liq} = 120.128 - 1.255 \times TEMP + 0.008 \times TEMP^2 + 1.138 \times \delta^2H_{vap}$);

vii) statistical analysis: Goodness of estimation? Which parameters explain variation in estimated liquid $\delta$ values best?;

viii) Application of final equations to dataset, consequent check of isotope standards throughout measurement campaign using derived equations.

A procedure like this has several advantages: First, it uses additional information that might have influence on the measurements, such as clay and water content. Second, it incorporates these information into one procedure, namely a multi-
675 linear regression. Third, an extra calculation step for the vapor-liquid conversion that exists in several forms can be avoided. Finally, the derived relationships can be objectively assessed using goodness of fit measures, tested throughout the measurement period, and, if required, adapted later. Thus, we recommend this way of calibration and derivation of liquid water $\delta$ values for future studies. However, we would like to point out that there might be other considerations evolving 'along the way' and different opinions on how to best calibrate *in situ* data exist.

**4 Water isotope-enabled modeling of the soil-vegetation-atmosphere continuum – opportunities emerging from *in situ* measurements**

The movement of water in an ecosystem is often measured at specific points, e.g. transpiration of one or a few leaves, sap flow in one or a few trees, soil moisture at certain depths in a soil profile. This is also true for new approaches measuring water stable isotopes *in situ*; i.e. the limitations of destructive sampling in regard to spatial resolution remain (though portable probes
are existing that might remedy this situation). In order to obtain reliable estimates of the measured variables for a catchment or even an ecosystem, those point measurements have to be upscaled to a wider area. This can be done by transferring the observations made and the knowledge gained into mechanistic, physically based models (e.g. Crow et al., 2005). Models can

also help to identify the dominating processes that govern water fluxes and residence times across the soil-plant-atmosphere continuum and are used to investigate subsurface processes that cannot be measured easily like root water uptake, preferential flow as well as percolation and mixing of soil water and groundwater recharge (Sprenger et al. 2016). A better mechanistic understanding and parametrization of these hydrological processes will in turn benefit models across scales – from field sites (e.g. Sprenger et al., 2015) to catchments (e.g. Birkel et al., 2014) up to global scale Earth System Models (Clark et al., 2015). At the catchment scale, tracer-aided modeling has become a significant research topic due to the higher availability of datasets on water stable isotopes measured in precipitation and streamflow (Birkel and Soulsby, 2015). By adding a travel time component, these approaches enable a combined representation of water velocity and celerity and ultimately allow to better represent ecosystem solute transport and get the right model output for the right reasons (McDonnell and Beven, 2014). It was shown that incorporating soil water isotope data into rainfall-runoff modeling improved the identifiability of parameters when simulating the stream water isotope composition (Birkel et al. 2014). However, Knighton et al. (2017) point out that in some catchments, isotope variation of streamflow might not react strongly to vadose zone ecohydrological processes and depending on the research question, model performance should be evaluated also including a comparison of modeled and measured soil water isotopes of the unsaturated soil. Furthermore, it is not clear how (isotopic) heterogeneity of soil and plant water values affect catchment-scale flux estimations, as such high-resolution measurements are just becoming available now. This illustrates the need for a better mechanistic understanding of sub-catchment processes and a concurrent comparison of model estimations and field measurements.

To address this, an increasing number of ecohydrological models were adapted in the last years to incorporate the movement of water stable isotopes between ecosystem water pools. The low temporal resolution that is usually associated with destructive sampling of water stable isotopes as compared to other soil physical and plant physiological measurements (e.g. soil moisture, matric potential, sap flow) limited their application in the past (Meunier et al., 2018). The continued and more in-depth observation of water stable isotopes in vadose zone water pools and plant water uptake will hence likely provoke the addition and revision of ecohydrological processes in isotope-enabled land surface models (Stumpp et al., 2018).

Table 2 summarises physically-based models that are able to simulate water movement and the water stable isotope composition in different ecosystem water pools and specifically, different depths of the vadose zone and/or plant water. As presented *in situ* approaches measure the water stable isotope composition in field studies with a certain level of limited spatial resolution, we focus on process models on the plot to catchment scale and spare listing isotope-enabled land surface models. We also include applications of the respective models that focus on investigating water fluxes and their isotope dynamics. A detailed description and comparison of listed models is beyond the scope of this review. Rather we want to illustrate the broad variety of options and benefits from incorporating water stable isotope data collected *in situ* in plant water and across soil profiles into isotope-enabled ecohydrological models. We further aim to encourage collaborations between field scientists and modelers. Both field measurements as well as modeling approaches are becoming increasingly complex and require substantial training and experience. Conclusively, it might be unrealistic to have both carried out by the same person. In addition, modelers and field scientists often speak 'a different language', i.e. look at processes from different angles. We therefore would like to

stress here that increased collaboration is inevitable. This might also include publication of 'cleaned' datasets and offering them to the community, as it is common in other disciplines.

*Selected examples for including isotope data into modeling studies*

Observed differences of isotope composition of bulk soil and mobile water and current discussions on the two-water world hypothesis, motivated Sprenger et al. (2018) to incorporate two soil pore domains, i.e. mobile and bulk soil water, into vadose zone modeling. They showed that accounting for both slow and fast water flow components with differing isotope composition and isotopic exchange via water vapor improved the simulation of soil water isotope dynamics. Also focusing on isotopic

effects on soil water, Rothfuss et al. (2012) used data from a controlled monolith experiment to calibrate SiSPAT-Isotope with measured soil volumetric water content and $\delta$ values across soil depths and in plant material to better understand the processes controlling evapotranspiration partitioning. They emphasize the importance of correctly determining the kinetic fractionation factor and the depths and isotope composition at the soil evaporation front and deduct recommendations on the location of measurement points when partitioning evapotranspiration in the field.

To advance the understanding of root water uptake and specifically assess the age of water used by two tree species (*Picea abies* and *Fagus sylvatica*), Brinkmann et al. (2018) used HYDRUS-1D and a set of water stable isotope data across soil depths and in plant xylem. They showed that temperate trees not only rely on recent precipitation but that even precipitation from the previous year substantially contributed to tree water supply (see Fig.1, lag-time). While also focusing on one single plant Meunier et al. (2017) used a 3D root system in a fully mechanistic soil-plant model (R-SWMS) to increase the realism and

potential for improved process-understanding of root water uptake. By comparing measurements of soil physical parameters and $\delta$ values with modeling results, the authors verified the concept of hydraulic lift and were able to quantify the amount of water released into the soil by the root system. Their simulation suggested that the magnitude of this water release by roots is controlled by two factors, root radial conductivity and soil hydraulic conductivity.

On the catchment scale, Knighton et al. (2020) used xylem isotopes (seasonal resolution) and soil water isotopes (weekly

resolution) in the fully distributed model EcH2O-iso to investigate the importance of tree water storage and mixing. When including this storage component, they found a better agreement between simulated and observed $\delta$ values of xylem water for summer and fall. They conclude that considering storage and internal mixing is likely advantageous when using isotope composition of xylem water not only in physically-based ecosystem models but also in statistical models calculating root water uptake depths.

While the models and applications described above investigate water movement at the plot and catchment scale, water stable isotopes are also included in multiple land surface models, e.g. iCLM4 (Wong et al., 2017), ECHAM5-JSBACH-wiso (Haese et al., 2013), Iso-MATSIRO (Yoshimura et al., 2006), NASA-GISS ModelE (Aleinov and Schmidt, 2006), ORCHIDEE (Risi et al. 2016), that can be coupled to atmospheric general circulation models (e.g. Risi et al., 2016). If model parts function as stand-alone applications to test particular ecohydrological processes (e.g. soil evaporation or root water uptake) but can also

be integrated into larger scale models, that combine modules that describe different water fluxes between system components,

the effect of one particular process on the whole system can be observed. By coupling the 1D model Soil-Litter-Iso to a land surface model, Haverd and Cuntz, (2010) demonstrated the importance of including a litter component into the model to better reproduce the evapotranspiration flux and its isotope composition at a forested site in Australia. Risi et al. (2016) performed sensitivity tests to the ORCHIDEE land surface models parameters to identify the potential of using water stable isotope measurements to better represent ecohydrological processes. They conclude that to best inform their type of model, water stable isotopes should concurrently be sampled in all ecosystem water pools. The authors point out that soil water isotope vertical variations are important to investigate and improve the realistic representation of infiltration pathways.

In contrast to physically-based models that aim at realistically describing physical processes of water and energy fluxes over time with mathematical equations and usually need substantial computing power, conceptual models are less complex and faster due to their spatial integration but rely on calibration parameters reducing their physical realism (Asadollahi et al., 2020). StorAge Selection functions are a recent approach combining water flow and transport processes to represent the effect of storage and biogeochemical processes on the water age distribution of catchment outflow (Rinaldo et al., 2015). Asadollahi et al. (2020) used water stable isotope data of lysimeter experiments to compare this approach with physically-based HYDRUS-1D simulations. They explain similarities and differences between modeled lysimeter drainage and evapotranspiration and discuss age dynamics of different water fluxes. Taking advantage of the high temporal resolution of *in situ* data of the isotope composition of xylem water isotope, StorAge Selection functions could also be used to investigate the importance and the effect of tree water storage and internal mixing on the $\delta$ values of xylem and water age of sap flow (Matthias Sprenger, personal communication).

Concurrent measurements of the water stable isotope composition in plant xylem and potential plant water sources, like different soil depths and groundwater, is an indispensable approach to determine root water uptake patterns and the relative contribution of present water sources. Rothfuss and Javaux (2017) reviewed different methods to determine root water uptake depths. Most commonly, purely statistical approaches (i.e. mixing models) are used. While these can also benefit from a better representation of the temporal variability enabled by *in situ* measurements (Kühnhammer et al. 2020), efforts should be directed at using physically-based models. Those models, only accounting for 4 % of reviewed studies (Rothfuss and Javaux, 2017), enable a better mechanistic understanding of root water uptake and help to improve its representation in land surface models. These examples show numerous ways in which water stable isotopes as tracers of ecosystem water fluxes can be used to evaluate and improve physically-based soil-vegetation-atmosphere models. On the other hand, modeling approaches provide a more integrated (spatially and temporally) view on water fluxes and can inform field scientists by optimizing sampling in respect to its timing, temporal and spatial resolution, as well as identifying compartments and fluxes that play a critical role in the specific investigated ecosystem. Key challenges will be how to deal with natural heterogeneity across different scales and ecosystem water pools in order to correctly upscale *in situ* point measurements (Penna et al., 2018). Furthermore, accounting for temporal dynamics of water stable isotopes measured in different ecosystem compartments, i.e. soils, plants and atmosphere into only one model might require to incorporate a lot more processes and parameters and therefore potentially decreases parameter identifiability. It is however important to address these issues and explore the use of new *in situ* data to improve the

physically-based representation and parametrization of key ecohydrological processes on the local scale in order to improve predictions of large-scale models.

**Table 2**

**5 Summary and Outlook**

The goal of this review was to summarize the current state of *in situ* approaches for measuring and modeling the water stable isotope composition of soil water, evaporation and plant water (in both xylem and leaf transpiration) and point out current issues and challenges. We further aimed to provide a hands-on guide on basic principles and difficulties associated with applying *in situ* methods. Based on this, we propose to combine applications of *in situ* investigations in different compartments of the soil-plant-atmosphere continuum in the future.

*In situ* measurements are an inevitable step for any holistic study within the critical zone. The current design of many ecohydrological studies is still based on destructive sampling at discrete points in time and space. The number of artefacts (potential isotope effects) and methodological constraints (limited spatiotemporal resolution, issues of measuring different water pools with different extraction methods) associated with that (refer to introduction) is increasingly questioning established methodologies. While certainly - apart from advancements in *in situ* methods - new protocols for destructive 805 sampling and analysis are needed in order to account for the findings of the last decade, *in situ* methods provide an elegant way of overcoming a number of current limitations. For instance, the water pools measured in soils and plants using *in situ* methods are ultimately the same, i.e. the mobile fraction that actively takes part in water fluxes and exchange. Using any extraction method, the risk of extracting and comparing different water pools is high (an extraction temperature of 105°C, for example, will remove almost all water from a sand soil, but leave a notable amount in a clay-rich sample).

Another example is the high temporal resolution that can be achieved with such measurements which resolves the issue of lag time and enables the investigation of non-steady state conditions. Hence, *in situ* methods will be highly useful for any study involving rapid changes of environmental conditions, e.g. root water uptake studies, water partitioning, night-time transpiration, etc. They will also benefit long-term studies, such as monitoring combined reaction of soils and plants to droughts or extreme events. Moreover, high frequency *in situ* monitoring can elevate tracing of the water cycle via isotopic labeling 815 ($^2H_2O$ or $H_2^{18}O$) to a new level and will lead to improved parameterization for a novel generation of physically-based models. The same is true for isotopic mixing models, which currently follow the beforementioned 'shotgun' approach (Berry et al., 2018). Another aspect that can be studied in much greater detail than before is the process of hydraulic redistribution (Burgess et al., 1998), to name one. Combined with labeling approaches, it might be possible to quantify its relevance and impact on a much greater spatiotemporal scale. Let alone these examples, *in situ* methods comprise immense potential for future 820 applications.

Having that said, it should always be carefully evaluated, if an *in situ* approach is required for the purpose of the study - or if destructive sampling is sufficient. When carrying out *in situ* studies, the aim of the study determines the design of the system to be used and a good starting point would be to clarify the following aspects:

i) Is the particular study a long-term study (weeks to months) or rather short-term (days)?

ii) Is the goal to obtain data in a high temporal or spatial resolution (or both)?

This aspects aims to define if the system needs to be portable or rather stationary.

iii) Is it a tracer experiment or is the goal to obtain natural abundances of soil/plant water isotopes?

The setup of any *in situ* system is neither simple nor easy; stand-alone or even plug-and-play approaches are still not available. In order to obtain reliable isotope data, daily maintenance and troubleshooting is inevitable at present. **Developing an**

**automated, portable system** including isotope standard measurements, probes, valve systems, mass-flow-controllers, temperature controls etc. that requires less maintenance is highly desirable. The complicated technical setup and calibration process as well as the vast amount of data created which needs to be processed carefully might be a reason why only a few research groups have conducted *in situ* studies so far. We hope to shed light on some of the technical aspects involved and clarify those through this review.

Despite the abovementioned issues, ***in situ* approaches for monitoring depth-dependent soil water isotopes** employing gas permeable probes have advanced tremendously in recent years. It now seems feasible to obtain measurements of natural abundances of soil water isotopes in a high temporal frequency. For **monitoring the isotope composition of xylem water *in situ*** under field conditions, on the other hand, there is only one existing study applying isotopic labeling (Volkmann et al., 2016a). Future efforts should be directed towards testing and improving the methods suggested and develop novel approaches

with the ultimate goal to measure natural abundances of plant water isotopes *in situ* (Beyer et al., 2019; Kühnhammer et al., 2020; Marshall et al., 2020). Subsequently, continuous soil and plant water isotope measurements should be combined (for a recent example, see Orlowski et al., 2020). Chamber-based measurements of transpired and evaporated water vapor are well established and have mainly been employed in frameworks focusing on partitioning of ecosystem evapotranspiration (Dubbert et al., 2013, 2014a, 2014b; Rothfuss et al., 2012) or studying isotopic fractionation during soil evaporation (Or et al., 2013)

and the leaf water isotope composition (Cernusak et al., 2016; Song et al., 2013, 2015a; Wu et al., 2013). They have also been used in ecohydrological studies tackling questions, such as root water uptake depths (e.g. Volkmann et al., 2016a). However, given the critical and often violated assumptions of isotopic steady-state of transpiration (i.e. $\delta$ values of transpired vapor not equal to that of xylem water; see e.g. Dubbert et al. (2014b); Piayda et al., (2017); Simonin et al. (2013)), this can be difficult under natural ambient conditions.

Despite the great advances in monitoring depth-dependent soil water isotopes *in situ*, there is no generally accepted calibration protocol existing yet (such as van Geldern and Barth, 2012 for water samples). Hence, **homogenization of calibration and validation protocols** are required. We propose here to make such a development based on the ideas of Oerter et al. (2017), which is – in the authors' opinion - the most complete of all currently existing approaches. It also provides an objective way of handling the data (via statistical measures) and is very flexible in including/excluding additional factors that might be

relevant (e.g. mineral-mediated fractionation). In terms of calibration, we further suggest that laboratory standards are provided using the same media that is to be measured (e.g. use standards prepared and measured in soils when measuring soils *in situ*) in order to fulfil the assumption of identical treatment principle, which has been violated in a number of studies. We contacted the authors of the original bag equilibration method (Wassenaar et al., 2008) with this question and obtained the following response: *'We and others have wrestled with this and you are correct the original publication is technically not an identical treatment. I suppose the real question is how much does either approach matter in practice vs its convenience – are we talking only 10th's of a permil bias (maybe not an issue) or a lot more (worrisome)?"* They also noted that *"it is also not identical treatment if you dry and wet soil or sand with lab standard waters, as some soils may have more potential for bound residual water or isotope exchange with clay particles, for example, or if the soil standard properties differ a lot from field samples."* (L.I. Wassenaar, personal communication). For this reason, an ideal preparation of soil standards is not existing at present. However, running pre-*in situ* laboratory tests using soil from different depths (e.g. A and B-horizon) from the site to be measured, oven-dry it, spike it with different water contents and measure it over a range of temperatures and water vapor concentrations will give a sound baseline for calibrating the on-site data. For the field calibration, soil standards (e.g. two to three) for each soil horizon should then be prepared and measured for each sequence in the field. We further propose to install TDR probes in each of the standard bags to keep track of the water content and temperature which is needed for the calibration.

For **validation**, it has been shown that a comparison of cryogenically extracted samples, although this has been the standard method for decades, with equilibration methods is not feasible for soil samples because different water pools are measured with the two approaches. The same might be true for plant samples. There is an urgent need to develop alternative ideas. For soils, a comparison of *in situ* data with destructive sampling and using the bag equilibration method might be a way. However, the issue of spatial heterogeneity between the two measures remains. For plants, the bag equilibration method might also be feasible but has not been tested thoroughly.

For both soils (e.g. the upper soil layers) and plants, the effect of **organic contaminants** (such as volatile organic compounds - VOC) on *in situ* measurements needs to be evaluated and measures developed to correct for it during post processing. Such might be included into the multi-step procedure suggested by Oerter et al. (2017). As stated, a method for correcting liquid water samples for the influence of organic substances already exists (Lin et al., 2019; Schultz et al., 2011; Wu et al., 2013) and could be easily adopted to vapor-phase measurements. However, it needs to be determined before if contamination even plays a role for data obtained *in situ*.

Another recommendation related to data treatment is the establishment of a way of **evaluation if equilibrium conditions prevailed** at the site of isotope exchange during the *in situ* measurement (e.g. inside of the gas permeable soil/tree probe). All reviewed studies presented herein use some sort of equilibrium  vapor-liquid conversion (e.g. Horita et al., 2008; Majoube, 1971). Only one of them (Marshall et al., 2020) evaluated if this assumption actually was true for their particular setup (flow rate, exchange length, etc.).

To estimate relative humidity (per definition the ratio of actual to saturated water vapor pressures) in boreholes the ratio of "water vapor concentration" (in ppmv) which is directly measured by the laser spectrometer can be compared to the saturated

specific water vapor concentration at stem temperature T (measured using a thermocouple or PT100 sensor). If these two

(roughly) match, it is likely that the chosen parameters of the (physical) system are suitable to confirm the assumption of equilibrium conditions. It also reveals potential for condensation under the given environmental conditions. Ideally, relative humidity $h$ should approach 1.0. Marshall et al. (2020) used this approach and stated that for values of $h$ substantially lower than 0.8, the assumption of isotopic equilibrium might be violated. In simple words, this would mean that the flow rate chosen is too high to allow for isotopic equilibration during the passage time through the stem borehole (or membranes used for soil

water isotope measurements). Hence, the flow rate would need to be lowered. We recommend for any system to check $h$ for evaluating if the defined settings of the physical setup are suitable. This concept is applicable to both push-through and pull-only setups (but if additional dry air is introduced to lower the water vapor concentration directed to the laser spectrometer this needs to be included into the calculations).

One might argue that via equal treatment principal, saturation is theoretically not necessary because it can be accounted for

during calibration. However, this would require, for instance for soil samples, a preparation of soil standards with the *exact* same conditions as at the measured soil depth (water content, temperature), which is practically not feasible.

In a concluding chapter, we propose a combined soil-plant *in situ* monitoring system which - in the authors' opinion - represents a holistic way of investigating dynamic ecohydrological processes at the interfaces of soil, vegetation and atmosphere.

**One system, one methodology – A call for combined *in situ* studies**

The authors of this study have been involved into the development of *in situ* methods for nearly a decade. Based on this literature review and their own experiences, an 'ideal' system is presented in Fig. 7.

**Figure 7**

The – admittedly highly complex – system depicted in Fig 7. combines measurements of all compartments covered in this

review. A setup like this would enable one to monitor the complete cycling of water through soils and plants: i) gas permeable probes for measuring depth-dependent soil water isotope ratios (supported by soil moisture/temp. sensors for the equilibrium calculations); ii) soil chambers for monitoring the isotope composition of evaporation; iii) stem probes or stem boreholes (supported by thermocouples for the equilibrium calculations); iv) leaf chambers for monitoring the isotope composition of transpiration and finally the monitoring of atmospheric water vapor. Ideally, these fluxes are all controlled by one

valve/manifold system. Through the inlet of each measurement stream, dry air with the required flow rate (MFC 1) can be directed through the probes/chambers. At the same time, it can be used to flush the systems prior to the measurement sequence with dry air (diving air, synthetic air or $N_2$ for removal of condensed water in the lines). The equilibrated water vapor then is sent back through the manifold and to the laser spectrometer. A second mass flow controller (MFC 2) offers the opportunity to dilute the sampling air if the water vapor concentrations are too high (less precise values above certain water vapor

concentration threshold). Any connection is an opportunity for leaks. The system is therefore limited to as little connection pieces as possible (i.e. one piece of Teflon tubing or stainless-steel capillary from the probe/chamber to valve

system/manifold). The excess tube avoids the possibility of overpressure at the analyzer inlet. The calibration unit consists of a user-selected number of soil standards for the soil measurements and water standards for the plant water isotope measurements. Additional (optional) components might include a higher number of monitored trees and/or soil profiles
(heterogeneity), sap flow probes, stem water content sensors and for the soils, matric potential and soil moisture content sensors. Though the depicted setup is constructed as push-through system (dry air is pushed through the compartments to be measured and equilibrated therein), it can be operated in pull-only mode as well.

When reading through this explanation, the reader probably gets the impression that this is very complicated. Admittedly, it is; and despite its complexity critical minds might still request if the suggested procedure is a true identical treatment. However,
a holistic approach for all relevant isofluxes would have an enormous potential for improving process-understanding (e.g. travel times, water sourcing, fractionation, storage times) and isotope-enabled modeling.

It is, thus, the task of the community to further improve, but also simplify *in situ* measurements. We encourage the community to carry out and test *in situ* systems. The increased technical effort for the setup is often compensated by far with the higher spatial (if using probes as mobile version) and temporal resolution.

Lastly, it needs to be clear to anybody applying *in situ* methods that a higher uncertainty has to be expected when working with such methods. While future efforts should certainly be directed to decrease those uncertainties as much as possible, it is equally important to communicate those uncertainties. Many of the 'old' studies are employing a very low number of samples, for instance for plant source water studies. They often end up with strong statements, but completely neglect the dynamic character of natural systems. Thus, only a (perhaps very small and biased) part of the story is reported. In order to improve the
understanding of ecohydrological processes it is inevitable to develop ready-to-use *in situ* monitoring systems; it is crucial for the community to further develop such methods and make them accessible to a larger group of researchers and practitioners in the near future.

**Acknowledgements**

This research was funded by the Volkswagen Foundation under contract number A122505 (ref. 92889). We thank Y. Rothfuss,
E. Oerter and one anonymous reviewer for their constructive and helpful coments during the review process and C.Stumpp for editing. We kindly appreciate additional input and ideas on the manuscript by L.I. Wassenaar, M. Sprenger and L. Kleine. Their advice and response to specific requests helped to improve this manuscript and make this a useful contribution.

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

**Funding Information**

This research is funded by the Volkswagen Foundation under contract number A122505 (ref. 92889).

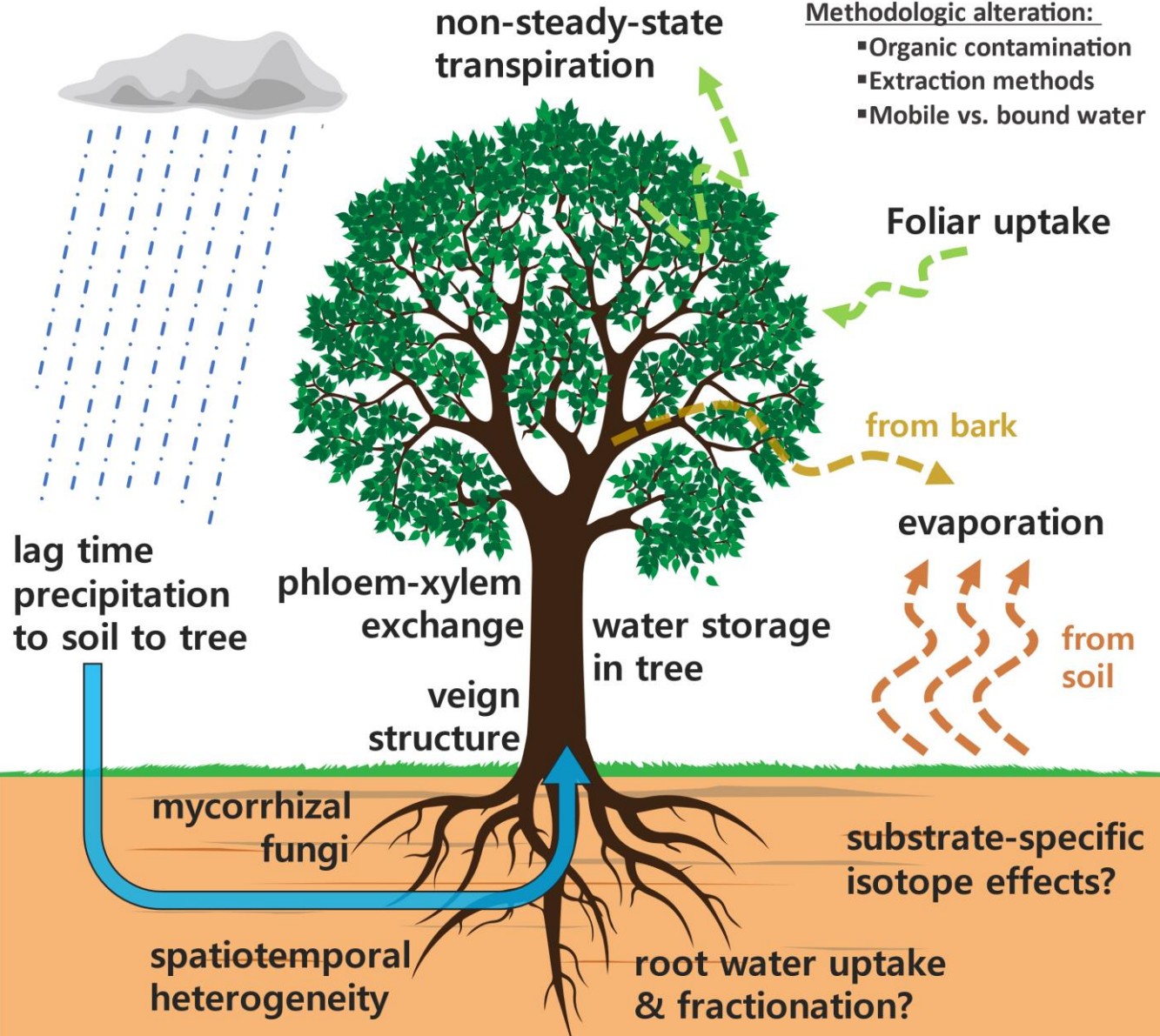

**Figure 1: A compilation of isotope effects potentially affecting the soil and plant water isotope composition**

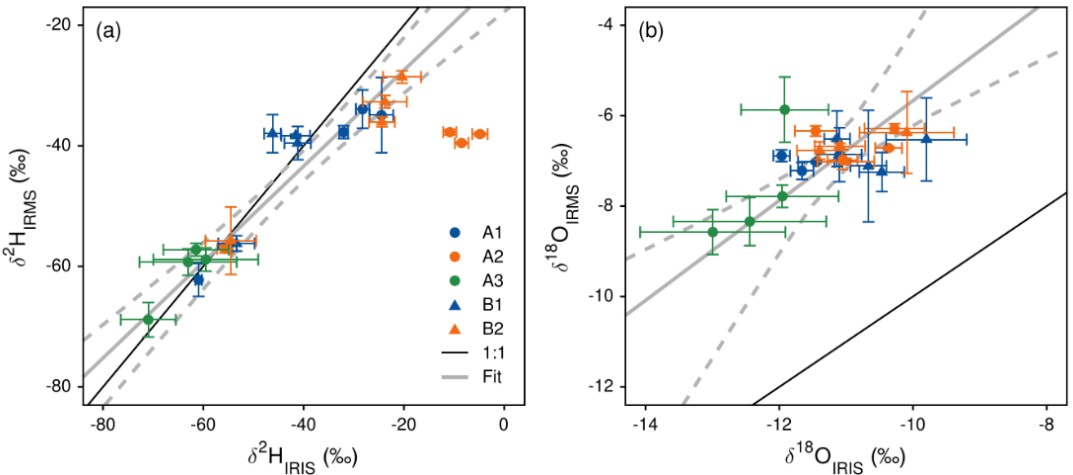

**Figure 2:** *In situ* **measured xylem water isotopes ($\delta^2H_{IRIS}$) and comparison to results obtained by cryogenic vacuum extraction after destructive sampling and measurement with mass spectrometry ($\delta^2H_{IRMS}$). Reprinted with permission from Volkmann et al. (2016b)**

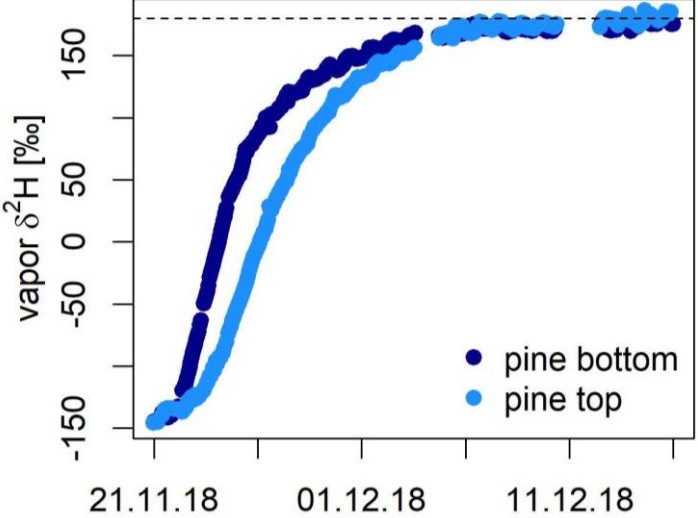

**Figure 3:** **Xylem water $\delta^2H$ values measured in the stem boreholes during a greenhouse experiment in Freiburg, Germany, on a pine tree (Marshall et al., 2020). Two boreholes were drilled through the stem and their $\delta$ values monitored over a period of two months. For both boreholes a close agreement of $\delta^2H$ between source water and *in situ* data was achieved.**

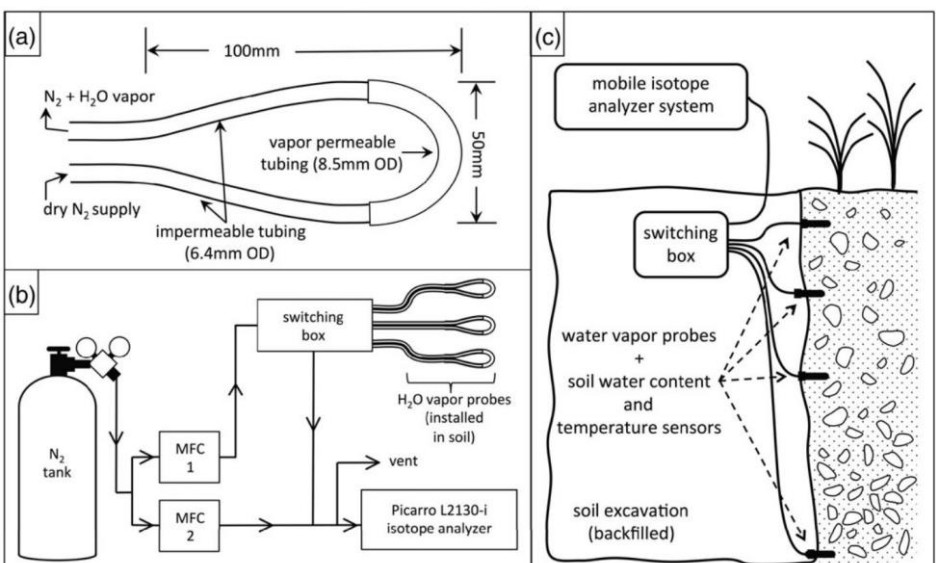

**Figure 4: Schematic drawings showing (a) an *in situ* soil probe consisting of a gas-permeable membrane and attached tubing; (b) a concept of the water vapor probe analytical system; (c) the field installation of an in situ system with additional sensors for recording soil moisture and temperature. MFC = mass flow controller. Reprinted with permission from Oerter and Bowen (2017). Note that different probe designs exist, and this is only one example.**

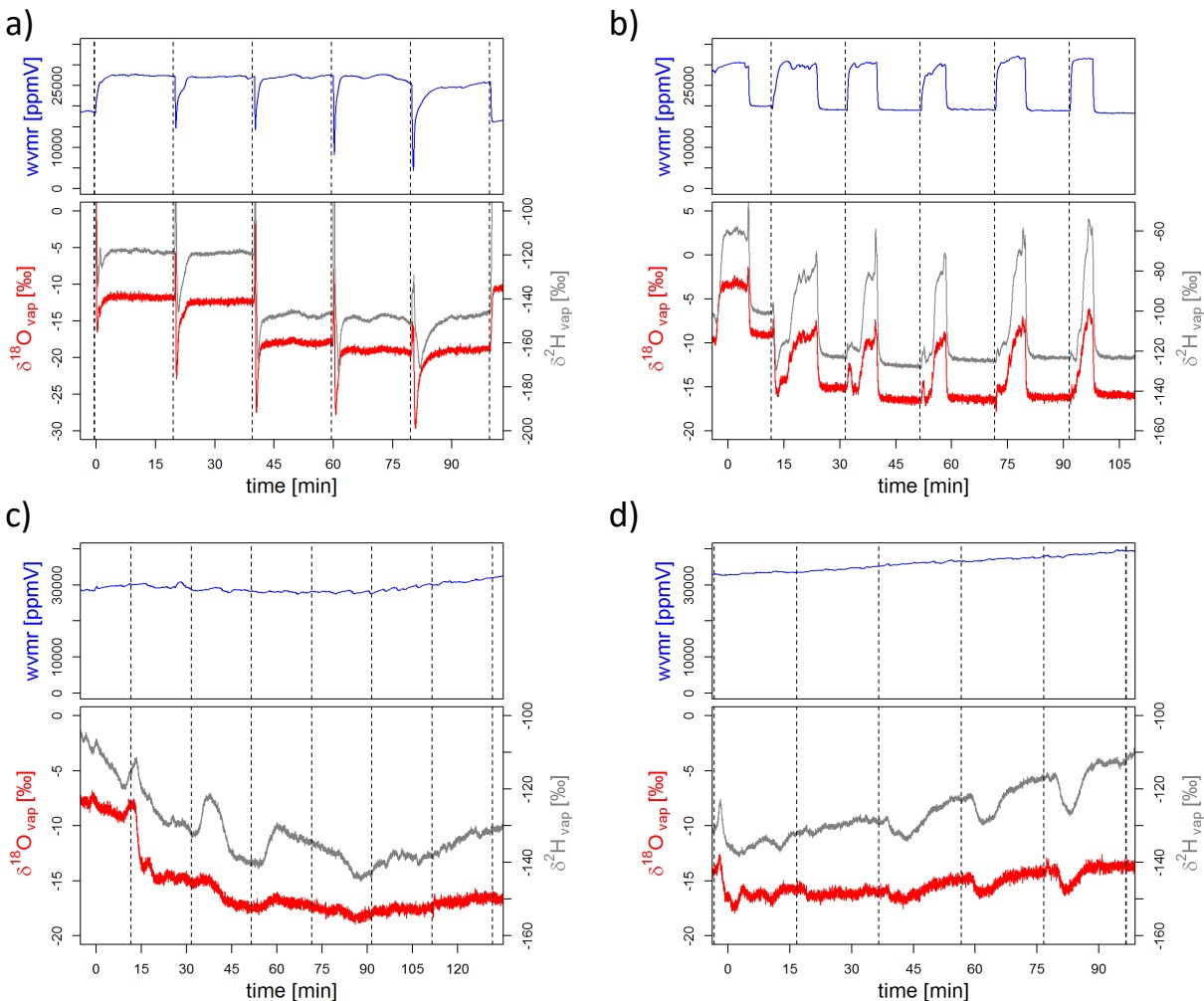

**Figure 5: Measurement cycle of an *in situ* system switching through different probes. Shown are water vapor concentration (vwmr in ppm) and the raw vapor values for $\delta^2H$ and $\delta^{18}O$ in permille. Each probe was measured for 15 minutes, then the manifold switched to the next probe (indicated by dashed vertical lines). The different panels show a) a clean measurement with a stable plateau for the three variables; b) a measurement where small amounts of condensation were present in the system, but then removed during the measurement phase resulting in a stable plateau towards the end of each cycle; c) and d) two examples of erroneous measurements, where condensation (=very high ppm values) does not allow the laser analyzer to reach a stable plateau.**

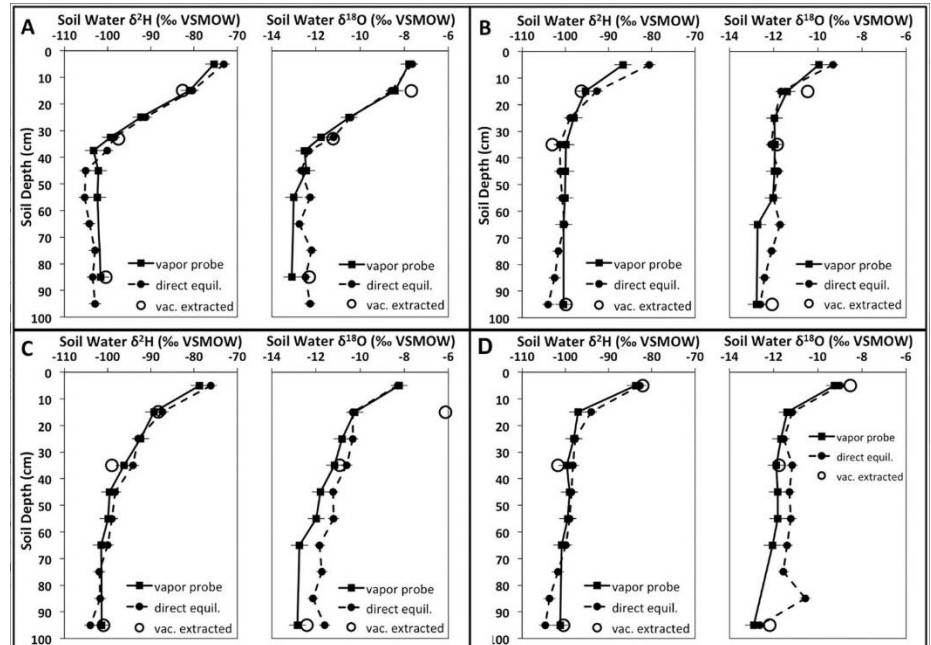

Figure 6: Comparison of soil water $\delta^2H$ and $\delta^{18}O$ values determined with the soil probes (solid squares, solid line), direct vapor equilibration (or: bag equilibration, solid circles, dashed line), and vacuum-extracted soil water (empty circles), with soil depth for four different sites. Analytical uncertainty in each vapor measurement methodology is denoted by horizontal whisker marks. Reprinted with permission from (Oerter et al., 2017).

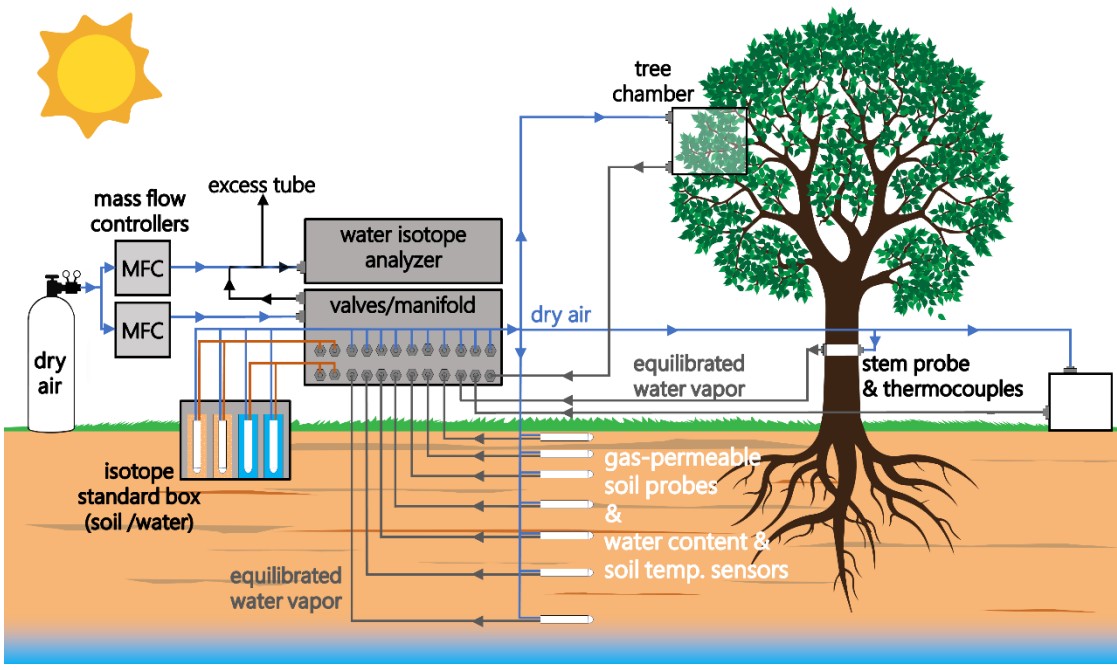

Figure 7: An idealized, yet complicated *in situ* system depicting all relevant components for a complete measurement of water isotopes of soils (depth-dependent and bulk soil) and plants (in tree xylem and at the leaf level).

