# Peer review of "In situ measurements of soil and plant water isotopes: A review of approaches, practical considerations and a vision for the future"

_Hydrology and Earth System Sciences, 2019_

## Short Comment (SC1) · 18 Dec 2019

This is a very nice review of the current state of the art for in situ soil and plant water stable isotope analysis techniques, and I applaud the authors for this timely contribution. The focus of the review is on laser-based techniques, and this is the route that most researchers currently take. However, the use of soil gas measurements taken via soil gas wells or probes, specifically $CO_2$, as a way to measure soil water oxygen stable isotopic composition, has not been included.

The basic principle is that in soils, the molar abundance of oxygen in soil water is orders of magnitude greater than the molar abundance of oxygen in coexisting soil CO2, so that the CO2 comes into oxygen isotope equilibrium with that of the soil water. Aliquots of soil gas are sampled via small diameter wells and stored in septum-capped vials for transport back to the lab. In the lab, the $\delta$18O of the CO2 is measured via continuous flow or dual inlet IRMS.

I suggest the inclusion of this technique in the review manuscript because it enables researchers without access to laser instruments, but with access to conventional IRMS instruments to be able to make in situ soil water O isotope measurements. Below, I include some notes on potential references that the authors may wish to include in their review.

Stern and colleagues laid the foundation for the O isotope relationship between soil CO2 and soil water:

Stern, L., Baisden, W. T., & Amundson, R. (1999). Processes controlling the oxygen isotope ratio of soil CO2: Analytic and numerical modeling. Geochimica et Cosmochimica Acta, 63(6), 799-814.

Breecker and Sharp developed a technique for the sampling of in situ soil CO2 and its isotopic analysis: Breecker, D., & Sharp, Z. D. (2008). A field and laboratory method for monitoring the concentration and isotopic composition of soil CO2. Rapid Communications in Mass Spectrometry, 22(4), 449-454.

Breecker and colleagues deployed their in situ soil CO2 system and successfully measured soil water isotope profiles at high depth resolution ($\sim$10cm depth increments) at monthly intervals for more than a year at several sites:

Breecker, D. O., Sharp, Z. D., & McFadden, L. D. (2009). Seasonal bias in the formation and stable isotopic composition of pedogenic carbonate in modern soils from central New Mexico, USA. Geological Society of America Bulletin, 121(3-4), 630-640.

[Figure]

Oerter and Amundson measured soil water O isotope composition via soil CO2 at four sites for over a year. They were able to develop estimates of the depth of plant water uptake via comparison with xylem water samples, thus establishing the ecohydrologic applications of the soil CO2-water technique:

Oerter, E. J., & Amundson, R. (2016). Climate controls on spatial and temporal variations in the formation of pedogenic carbonate in the western Great Basin of North America. GSA Bulletin, 128(7-8), 1095-1104.

An added value of the use of soil CO2 sampling to measure soil water O isotope composition, is that the soil gas samples can also be measured for CO2, CH4, NOx, as well as other trace gases abundance. The ecohydrologic benefits and implications of these types of measurements were demonstrated in a later paper by Oerter and colleagues from the same field sites as the 2016 paper above:

Oerter, E., Mills, J. V., Maurer, G. E., Lammers, L. N., & Amundson, R. (2018). Greenhouse Gas Production and Transport in Desert Soils of the Southwestern United States. Global Biogeochemical Cycles, 32(11), 1703-1717.

---

## Referee Comment (RC1) · Youri Rothfuss (Referee) · 20 Dec 2019

I am reviewing the manuscript entitled "X Water Worlds and how to investigate them: A review and future perspective on in situ measurements of water stable isotopes in soils and plants" by Matthias Beyer and Maren Dubbert under discussion for subsequent review in HESS.

First, I have to state for the record that I have collaborated with Dr. Maren Dubbert, which is why I show my identity here. I only accepted to review their manuscript be-

cause (1) of its format (review), which is much less subject to controversy than it would be for a research article and because (2) I have never heard of the manuscript until the editor Christine Stumpp contacted me. The authors will see that I provide a number of tough but fair comments!

The manuscript is an exhaustive and timely review on the subject of online non-destructive analysis of soil and plant water isotopic compositions. It is well written, easy to understand (at least to me) and well structured, exception made of Part 2.2, which does not fit the scope of the paper, i.e., the "current status of in situ measurements of water stable isotopes in soils and plants". (I suggest removing it in my specific comments). Also I leave to the editor to decide if the authors should extensively make reference to their own unpublished/non peer reviewed work for proving their point.

My first general comment is the following: the methods that we employ can only be as good as the understanding that we have of the processes in play. For this we have physically-based models, that, despite their limitations, compile our knowledge and propose possible explanations of (isotopic) observations. The "in situ" methods – I argue later that they should be renamed "online" method – induce a paradigm change in isotopic analysis and leave us with much more information to process. Effort in linking the data stream to existing isotope-enabled models is therefore crucial and should have its own section here, rather than mentioned now and then throughout the manuscript. I wrote together with Mathieu Javaux a paper on the specific subject of model-to-data exchange for the specific case of root water uptake analysis (Rothfuss and Javaux, 2017).

My second general comment is that, in my opinion, the two-water-worlds (TWW) hypothesis is not the development trigger of such techniques. I don't see these methods as a way to validate the hypothesis. I know that the TWW attracts a lot of attention therefore should be evocated in the main text but not named in the title, especially since you only mention it in the introduction. Furthermore my opinion is that we cannot investigate these water worlds (or pools, even though the link between these two concepts

is not obvious) on basis of water stable isotopic measurements: using them makes the analysis biased, as we implicitly recognize that these different worlds/pools exist. There are other (e.g., geophysical, combined tracer) techniques that work on different premises and should be used towards a proper characterization of these pools/worlds.

My third and last comment is about spatial (vertical and lateral) discretization and representativeness of the online methods. They still poorly compare with those of the destructive sampling approaches. What do we do in the field with these highly resolved and long-term isotope compositions data series if the information is only relevant at the square meter-scale or for this particular plant individual?

Looking forward to a revised manuscript!

Cheers -Youri Rothfuss

Technical comments:

Title: you do not measure the water stable isotopes, rather their isotopic compositions.

Abstract

L10. "involving water stable isotopic measurements"

L11. "e.g.", not "i.e."

L15. "in the soil-vegetation-atmosphere continuum" (or e.g. "at the soil-atmosphere interface" etc.)

L17. Mention and explain shortly before what for you constitutes a "water pool".

L17. Reformulate: you mean certainly "spatial variability and temporal dynamics of the water isotopic composition in terrestrial ecosystems".

L18ff. "in situ" is a vague term... destructive as well as non-destructive sampling are always done in situ. On the other hand, the measurements are performed on-line vs off-line. Consider using another terminology throughout the MS, e.g. the aforementioned

"on-line"

L19. "disentangle" is used twice in the abstract. Find alternate term here maybe. . .

L21. "water stable isotopic compositions".

L21-28. Should be one single paragraph.

L24. An interface cannot be threefold. You are talking of a continuum here. They are many interfaces present within a continuum and in this particular case three, i.e., the soil-root, soil-atmosphere, and plant-atmosphere interfaces.

L25. This is a given. Consider omitting.

L25-26. "In situ methods for soils are well established". The literature as well as yourself later say otherwise.

1. Introduction

L45-48. What is always omitted and should be stated here is the water demand vs water availability. Experts on root water uptake processes know that if a plant does not need to extract water that is "easily" accessible, because it e.g., does not transpire enough or is adapted in this regard. . .it will keep extracting "less available" water. "Easily accessible" water is also poorer in dissolved oxygen and nutrients and might be the last place to look for water for some species. This should be discussed.

L52. "at the soil-vegetation-atmosphere interface". See my previous comment.

L55-64. Earlier you are talking of water "worlds" but here of "pools". They are not the same. This should be discussed as well.

L62-63ff. Edit reference format. It should be "et al."

L63-64. "As a consequence, a big question arises: Are all source water studies wrong?": this is a scary statement (!) Also do not say "wrong", as there is a moral aspect to it. "biased" sounds about right.

1.1 Are ecohydrological source water studies biased? – The need for in situ methods

L70 "low suction tension" :)) Just say low "low tension" or "low soil water tension"

Fig.1. You are missing evaporation! Also phloem-xylem exchange, Péclet effect, and cuticular evaporation could be mentioned. In addition you should use arrows for fluxes rather that for lag time. Anyway, is this (nice) drawing really needed? It is not quite informative, and you don't make use of it in the text.

L79. "water isotope composition values"

L101. "2 Review: In situ approaches for measuring soil and plant water stable isotope composition"

L102. "2.1 In situ soil water isotope composition depth profiles"

L103-110. Nice §!

L105. $\delta$2H needs definition

L105-106. Take care of the scientific grammar / isotopic terminology: it should read "The determined isotope composition values agreed well with that of the samples extracted from the soil".

L103-212. "The concepts tested in Herbstritt et al. (2012) therefore can be seen as a baseline for all subsequent in situ soil water isotope studies.": I would not say this at all. Barbara Herbstritt's study is not on soil water, but on meteoritic waters. Their membrane as well as their modus operandi were not further used in the "subsequent in situ soil water isotope studies". This is also the case for Soderberg et al.'s study, which, even though it constitutes the first published work on online soil water vapor measurement, can be seen as an outlier as their method was not further applied. I see that they are two "families" of methods that rely on the same principles: those of Rothfuss et al. (2013) and Volkmann et al. (2014). The first one has seen to date more applications and further developments (Rothfuss et al. 2015; Gangi et al. 2015;

Quade et al. 2018; 2019, Oerter et al. 2017, 2019, Oerter and Bowen, 2017, 2019, Kühnhammer et al. 2019) that the second one (Gaj et al. 2016), certainly for the reason that the probes weren't at first off-the-shelf products (although Gaj et al. 2016 used already commercially available soil gas probes). We in Jülich personally did not have the idea to use membranes from Barbara Herbstritt's work, rather from previous applications in soil CO2, N2O, and CH4 sampling [Dinsmore et al., 2009; Hartmann et al., 2011; Neftel et al., 2000]. These studies could be a nice addup to this nice section. Also you are almost up to date: you are missing the following study:

Quade, M., et al. (2019). "In-situ Monitoring of Soil Water Isotopic Composition for Partitioning of Evapotranspiration During One Growing Season of Sugar Beet (Beta vulgaris)." Agricultural and Forest Meteorology 266–267: 53–64.

L158. "directed in"

L158-159. Dry air cannot equilibrate with some isotopic composition. Please reformulate

L169. "combined measurements of $\delta$18O and $\delta$2H" I presumed.

L213-300. Part 2.2 feels not to fit the scope of the paper, i.e., the "current status of in situ measurements of water stable isotopes in soils and plants". I suggest removing it. I will not distract the reader away from the core of the paper.

2.3 In situ measurements of plant xylem water isotopes

L312. "can be observed"

L312-313, 316ff etc. "xylem water isotope compositions"

L317-318. Define IRIS and IRMS, and "BSquare-weighted M-regression"

L320. Use ‰

L321-349. I am not sure that you should review non-published literature. . .I leave that

to the editor to decide.

L337 (goes as well for at L338). "$\delta$18O xylem isotopic composition values were lower than those of the source water"

L349. This table is central to your review and should be moved from the SI to the main text and named Table 1! It should be mentioned at the beginning of section 2, so that the readers can consult it while reading.

L350. "3 Setup, Calibration and Validation of in situ measurements of soil and plant water isotope compositions"

L354. "for sampling the water vapor"

L355-358. Exactly my specific comment from before on section 2.2 (+see my general comment)

3.1 Materials and approaches for sampling soil water vapor

L362-363. Only Volkmann and Weiler (2014) built probes in the pure sense. The rest of the authors (apart from Gaj et al., 2016) basically just cut sections of the gas-permeable tubing and took care that it was water tight when installed in the soil and properly connected to the sampling lines.

L365. "Self-made" is not appropriate. You mean tubing vs membranes inside sensor.

L367-392. This part echoes my comment from before regarding text at L103-212. I suggest a bit of streamlining and incorporating the one part (L103-212) into the other (L367-392) or vice versa.

Two comments here:

1- soil vapor is not necessarily water saturated for a soil liquid-water vapor equilibrium to be reached. Isotopic equilibrium happens between the two phases at relative humidity < 1 at high soil water tension.

2- there is no (and should not be!) overpressure inside the tubing, otherwise you push the dry air inside the soil and disturb the vapor-liquid coupled state. For this the sampling lines downstream of vapor collection should be kept as short as possible to minimize resistance to air flow and of course an open-split. But even with long collection lines, the increase of pressure is/should not be observed. You have, on the other hand an overflow (rather than overpressure).

L380. "access tube just before the analyzer inlet": the technical accepted term is "open-split"

L385-390. What is the point of having a slightly easier system to implement/use if in the end it might not provide reliable data? My opinion: make a decision as this will influence the readers and potential future users.

L390-391. You said this already above. Consider erasing.

3.2 Saturation of water vapor, condensation and dilution

L400-410. A bit of reorganizing would help: you need to distinguish between condensation prior sampling starts from condensation during measurements. To get rid of the first problem, you should do iii) and to avoid the second kind of problem you should do points i) and ii).

Ideally you need (i) a three way valve after collection point and before dilution and (ii) a two way "normally open" valve between dilution point and open-split. This former allows you to:

1) flush the liquid water out of the gas-permeable tubing and out of the downstream sampling line prior measurement [valve is turned on and act as open-split]

2) flush the content of the gas-permeable tubing to the laser-spectrometer [valve is turned off]

3) flush the sampling line after dilution point and to the laser spectrometer open-split

once sampling is complete to lower the water vapor mixing ratio in it and avoid condensation problems in between measurement phases [valve is turned off and act as open-split]. Valve (ii) is then closed to keep the sampling line dry and the laser spectrometer samples from its own open-split.

Finally for increased security, you shall heat the sampling lines from the soil/trunk surface to dilution point.

This could be incorporated into Figure 7.

L420. "Correction for water vapor concentration dependency of the laser spectrometer isotopic composition raw readings"

L426. "3.3.1 Soil water vapor isotope standards"

L427. Not only with the same medium but thanks to the exact same method: the standard water vapor has to be collected in the same manner as that of the soil, i.e.:

(i) pass through the walls of the same membrane product coming from the same production batch (e.g., tubing of the same age, thus having the same permeability to water vapor), be exposed to equivalent environment (pH, chemical conductivity of the soil solution).

(ii) at an equivalent flow rate as for the samples

Also not to forget: a soil is not only characterized by its natural texture, but also structure, so the soil water vapor standards should ideally have the same structure on top of the same texture as the investigated soil. In relation to this: in my group we however did not see an effect of soil water content on raw isotopic readings. In Quade et al. (2018), we argue that what Oerter et al. (2017) see is due – at least partly – to the fact that they test the membrane on unstructured soils.

L453-459. You should mention of course the work of Markus Schmidt and colleagues, however do not display their equations as they are specific to their instrument. Also

they used one of the very first L1102i (number 8 if I remember correctly :) and Picarro[®] (and this goes for other companies) have since greatly reduced the linear response of their instruments to water vapor activity.

What you should write, on the other hand, is a short description on how these dependencies should be investigated using, e.g. the aforementioned soil water vapor isotopic standards at different dilution rates.

3.3.3 Other corrections (mineral mediated fractionation, organic contamination, carrier gas and biogenic matrix effects)

L469-470. This relates to artificial unstructured soil mixtures...see my previous comment. We are dealing with...soils, so structure (bulk density for instance) might play a significant role!

L475-487. If this apply to soil and plant water online measurements, the associated effect on raw isotopic composition readings should be orders of magnitude lower than for extracted water after inverse sublimation of for evaporated water following pyrolysis... You could mention that this has not been investigated but you don't need to review the literature that deals with VOC contamination on liquid samples (out of scope).

In addition: what about the material out of which the sampling lines are made of and about their isotope effects? (there is also a dynamic adsorption-desorption equilibrium happening in them) Which one should we take - plastic or metal? Which kind of plastic, which kind of metal??

3.3.4 Drift correction

L494. Split "Conversion of vapor to liquid values" from "3.3.4 Drift correction". These are unrelated things.

L497-498. You can omit the book of Clark and Fritz (this goes as well for at L122): the (peer-reviewed) works of the great Juske Horita and Mustafa Majoube are enough :)

These equations refer then to another state of equilibrium, namely "static equilibrium". What we deal with here during purping is dynamic quasi isotopic-equilibrium. Rothfuss et al. (2013) showed that the purging and/or the PP material had an isotopic effect for 1H2H16O. And it is up to now the only real calibration available. These equations should be preferred over those of JH and MM or to the very least mentioned.

3.4 Validation –comparing apples and pears?

L538. Oerter et al. (2017) does not "propose a novel, innovative way of calibrating in situ data of soil water isotopes". You refer to Oerter and Bowen (2019), certainly.

L545-560. There is a good chance that such regression analyses will add more uncertainty to the measurements when not performed carefully.

Points (i)-(ii) omit the importance of the soil structure on soil water vapor flow. A note: the addition of water to a dry soil does not ensure a homogeneous soil water content.

Points (v)-(vi) apply to deconstructed soil samples, see previous comments

4 Summary and Outlook

L586-587. "we spare modelling approaches in this review, but refer to recent developments,...": see my general comment.

L618-621. In my opinion, this is not the way to go. It is much better to prepare different standards in an appropriate manner (repacked at the same field density) reflecting the different textures encountered in the field, so accounting for e.g. the effect of hydration and isotopic fractionation around clay particles.

It has to be proven if, indeed SWC plays a role when the soil is structured: my group has never seen such an effect (see Rothfuss et al. 2013, 2015; Quade et al. 2018; Kühnhammer et al. 2019) with compacted soil: this should mentioned/discussed.

L649-653. Rothfuss et al. (2013) did this...also do not cite a paper under review (Marshall et al. 2019).

L664-661. This is a very good idea! However the terminology used in a bit funky: the relative humidity is the ratio of actual to saturated water vapor pressures (not water vapor concentration) in the original definition. Also you certainly mean water vapor dry mixing ratio, which is typically expressed in vol-ppm. Why should the relative humidity be higher than 0.9 specifically? Look at the Kelvin equation and you'll see that for soil pF=7, relative humidity should yield to 0.97.

L671. Define compartments

L674. This only applies to trees, right..?

L689-693. Why would you say that? Scientist from other field might say otherwise...you can say, on the other hand, that online non-destructive methods are more complex to install/implement than destructive sampling ones. But as long as we do not make complex things complicated, we are on the safe side :)!

---

## Referee Comment (RC2) · Anonymous Referee #2 · 18 Feb 2020

The manuscript by Beyer & Dubbert reviews current methods of in situ measurements of soil and plant water isotopes. The methods are well described and an exhaustive review of past studies is given, before individual issues of current measurement techniques are discussed and finally suggestions for improvements are given. The manuscript is very well written and easy to read. The contents are valuable and worth publishing as the described methods are indeed still in development. Therefor, a comprehensive review such as this that collects the current knowledge is highly relevant. The title is slightly misleading however, as the manuscript barely deals with the "two or

x water worlds" hypothesis. I suggest to remove all parts before and including the ":" from the title and just call it "A review and future perspective on in situ measurements of water stable isotopes in soils and plants". I only have some minor comments for clarification or little bit more detail and suggestions for typos.

Specific comments L38: the more easily accessible water would be soil water that eventually becomes stream water, currently the sentence sounds like stream water is directly sampled L95: unclear if "leaf chamber" is a physical e.g. plastic chamber where leaves are studied or if stomatal openings, the sites of transpiration, are meant. L120: where's the difference of fractionation factors and equilibrium fractionation factors? Be a bite more specific what is exactly meant with the first term L164: which other value was used to linearly correct the measured hydrogen isotope values in vapor in the study of Rothfuass et al. 2013? Further, how did Rothfuss et al (2015) proof that their isotope values were reliable? They must have compared it to something. L173: according to Line 128, the first in-situ study in semiarid areas was by Sodeberg et al (2012) and not Gaj et al. (2016). Please clarify. L183: incomplete soil water extraction by which system? L200: since this system is the most complete, I am lacking a bit of information of what this system consists of exactly. Especially since later the authors of this manuscript suggest a more general, expanded approach to the cited study. The other methods were well described but here the description is a bit lacking. It seems to be a vapor-permeable membrane technique, but the advantage or difference to previously described methods is unclear. L210: I would not call this "trueness" but "accuracy". Since there is also a discussion ongoing of which water is exactly sampled by each method and which method can give "correct/true" values under which conditions. L408: method number 3 does not sound like a system to prevent condensation. The condensate between each flushing is just removed. Also for the example of Figure 5, where is the data coming from? Did you conduct this experiment or is it from another study? L445: the meaning of the sentence becomes unclear starting with "correction has mostly been applied [. . .]" L488: The description of carrier gas effects is very short compared to the rest L545: does the proposed method have to be repeated

for each individual study? I think that the final multiple regression results should be general and applicable for many studies. Please elaborate on this. L579: I do not see how the example of mobile water is related to the sentence before, that in situ methods overcome current method limitations.

Technical comments L61: citation: Hendry et al. 2015. As changes in citation style appears again in the manuscript, please keep it uniform everywhere. L83: sentence "and this is where..." should be shifted, as it currently reads very confusing. For example, "The scientific community agrees that one of the most important steps to investigate, disentangle, quantify and incorporate...." L87: delete the one "a" before "the same" L91: "kil lthe plant" please correct L92: Sentence sounds like time and costs of destructibe sampling result in larger sample amounts. L94: delete "of" L169: "O" for oxygen missing at the end of the sentence with "delta 18". Also unclear what combined measurement is referred to? I assume d18O in water and carbondioxide? L188: delete either "of" or "for" L200: "apllied" please correct L213: write already "isotopes" in the heading for 2.2 L252: "Gordon" missing L304: delete "and" L375: delete one "now" L415: "disappears" spelling L454: use "positive correlation" instead of "positive effect", as this sounds beneficial L465: "no" should be "not" L472: "A" should be "As [stated above]" L512: "is" should be "as" L514: do you mean "previous" instead of "succeeding"? L671: Fig 7 instead of Fig 6

---

## Author Comment (AC1) · 28 Mar 2020

We thank Erik Oerter very much for this positive feedback. It is very helpful to receive this comment by one of the (so far) few groups dealing with this topic. We are grateful for this suggestion and have considered to include a section on soil $CO_2$ measurements. Finally, we decided to include a short paragraph but would not like to extensively touch on the soil $CO_2$ measurements for the sake of not further extending this manuscript.

Kind regards, Matthias Beyer & Maren Dubbert

---

## Author Comment (AC2) · 28 Mar 2020

We thank reviewer#1 Youri Rothfuss for his constructive comments and suggestions. We considered each comment and you can find our replies below. We much appreciate the positive feedback and many well-founded points that certainly will help to improve the manuscript.

[1] The manuscript is an exhaustive and timely review on the subject of online nonde-structive analysis of soil and plant water isotopic compositions. It is well written, easy to

understand (at least to me) and well structured, exception made of Part 2.2, which does not fit the scope of the paper, i.e., the "current status of in situ measurements of water stable isotopes in soils and plants". (I suggest removing it in my specific comments).

Reply: Thank you very much for the positive feedback. We wrote this review in order to encourage people to further develop and simplify in situ methods and increase its application. The feedback from somebody who is amongst the people at the forefront of these developments helps us a lot. Part 2.2. We believe that it is important to also consider chamber based in-situ methods in this paper: 1) they are quite important for holistic approaches quantifying rapid soil-plant and plant-atmosphere feedbacks, 2) despite them being not as recently developed as the methods based on semi-permeable membrane, they are still quite novel and complete the "set" of in situ, field deployable observation techniques for plant and soil isotopic signatures. For example, we could gain better insight into water transport dynamics and storage/mixing in tree xylem by measuring at different points along the plant water uptake path, i.e. in soils, in tree xylem and in tree crowns (leaf chambers). Also, plant chambers are an important in situ tool to measure plant water uptake of herbaceous species/all species that don't have suberized stems/that don't have stems that are big enough to measure in the sap wood directly. Perhaps that chapter appears a bit disconnected from the rest because of different writing styles. We will consider rephrasing and homogenizing to create a better 'flow'.

[2] Also I leave to the editor to decide if the authors should extensively make reference to their own unpublished/non peer reviewed work for proving their point.

Reply: Both papers that this comment refers to are accepted and published meanwhile. We will update the reference list accordingly. (we were a bit counting on this while preparing the manuscript)

[3] My first general comment is the following: the methods that we employ can only be as good as the understanding that we have of the processes in play. For this we
have physically-based models, that, despite their limitations, compile our knowledge and propose possible explanations of (isotopic) observations. The "in situ" methods – I argue later that they should be renamed "online" method – induce a paradigm change in isotopic analysis and leave us with much more information to process. Effort in linking the data stream to existing isotope-enabled models is therefore crucial and should have its own section here, rather than mentioned now and then throughout the manuscript. I wrote together with Mathieu Javaux a paper on the specific subject of model-to-data exchange for the specific case of root water uptake analysis (Rothfuss and Javaux, 2017).

Reply: Thank you, this is a good point. We definitely agree that there should be modeling and measurement approaches going hand-in-hand. After careful consideration, we decided that we will add a section on isotope-enabled models and how in situ measurements relate to them. However, to do this, we would like to ask a third author to contribute in this regard (K. Kühnhammer). We leave it up to the editor to finally decide that.

[4] My second general comment is that, in my opinion, the two-water-worlds (TWW) hypothesis is not the development trigger of such techniques. I don't see these methods as a way to validate the hypothesis. I know that the TWW attracts a lot of attention therefore should be evocated in the main text but not named in the title, especially since you only mention it in the introduction. Furthermore my opinion is that we cannot investigate these water worlds (or pools, even though the link between these two concepts is not obvious) on basis of water stable isotopic measurements: using them makes the analysis biased, as we implicitly recognize that these different worlds/pools exist. There are other (e.g., geophysical, combined tracer) techniques that work on different premises and should be used towards a proper characterization of these pools/worlds.

Reply: We agree, the TWW are not the trigger for such techniques. Our personal opinion is that the TWW simply gives a new, fancy name to something that was already known before and is explainable by soil hydraulic relationships and basic soil physics.

[Figure]

This has been shown previously (Sprenger 2016 and Dubbert 2019). However, the TWW definitely triggered a more process-based interest when using water isotope data. The – slightly provocative – title we chose was meant to point to the fact that the explanation for the TWW might not be as simple as the proposers thought initially. To this end, the in situ, online, etc. (we will comment on that later) definitely help to disentangle such effects (e.g. lag time from soil to plants can be investigated now comprehensively, higher temporal resolution gives us a better idea of the methods accuracy, methodological problems -e.g. condensation- can be directly recognized in the field). Of course, it is not only in situ methods that will help in a deeper process-based understanding. It is true that it might not be necessary to mention the water worlds in the title. However, we'd like to emphasize again that our intention with this title was to also reach out to the two water worlds community, who might otherwise not be interested in the in situ topic at a first instance. In our opinion it is important to look at such hypothesis from another angle and a review should somehow account for that. Also, a comprehensive overview how in situ methods can help for a better process understanding, the processes need to be introduced first. The introductory section of a review paper is – in our opinion – a perfect platform for that. This is why we would like to keep Figure 1 as well (see later comment).

We propose the following for the revised version: - We would like to keep lines 39-54. - We propose to rephrase the title – if it is really necessary – to: 'A review and future perspective on in situ measurements of water stable isotopes in soils and plants'. However, we would appreciate an opinion from the editor regarding changing the title given our elaborations above. We believe that the current title is suitable for this review and approaches a wider community. Perhaps, we need to clarify better in a revised version why we chose the water world analogy.

[5] My third and last comment is about spatial (vertical and lateral) discretization and representativeness of the online methods. They still poorly compare with those of the destructive sampling approaches. What do we do in the field with these highly resolved

and long-term isotope compositions data series if the information is only relevant at the square meter-scale or for this particular plant individual?

Reply: This is a very good comment and we appreciate it. In the revised manuscript, we will elaborate on this aspect in Section 4 Summary and Outlook. It is certainly necessary to not only provide long-term and high-resolution datasets for one single plot. There are efforts to improve on that and several groups (e.g. the Freiburg, Utah and our own) are working on that. Having that said, many destructive sampling studies use a very limited number of discretization points (e.g. one or two soil depth profiles, a number of plant samples). In the end it depends on the purpose of the study how an experiment should be designed.

Technical comments:

[6] Title: you do not measure the water stable isotopes, rather their isotopic compositions.

Reply: Yes, this is a sloppy formulation. But we also do not measure isotopic compositions to be completely correct. (I checked recommendations by Coplen, 2011 and Bond & Hobson, 2012): Isotopic composition refers to a general observation based on isotopic information (e.g. "The isotopic composition of heron feathers varied between sites"). It should not be used to refer to relative differences in isotope ratios. And: 'Isotope ratios (R) are the simple ratio of the number of atoms of two isotopes in a material), and the $\delta$ value is a mathematical manipulation of a ratio of isotope ratios." Relative difference of isotope ratios" is cumbersome, and will not likely be universally adopted, so authors should use the term "values" or "$\delta$ values" (e.g. "$\delta$ values differed between years" or "the measured $\delta$ values are presented relative to the following international reference materials").' We will first clarify the terminology in a glossary (proposed by an additional anonymous reviewer via direct Email) and subsequently use either 'isotope values' or '$\delta$ values' throughout the manuscript (it is clear that the review is on water stable isotopes). 'Isotopic composition' is – in our opinion – a rather

uncommon notation.

Abtract

L10. "involving water stable isotopic measurements"; L11. "e.g.", not "i.e.";L15. "in the soil-vegetation-atmosphere continuum" (or e.g. "at the soil-atmosphere interface" etc.) Reply: Thank you. We will correct the above three. L17. Mention and explain shortly before what for you constitutes a "water pool". Reply: Thank you. As we got the advice to create a glossary of used terms in the beginning, we would incorporate 'water pool' into this. L17. Reformulate: you mean certainly "spatial variability and temporal dynamics of the water isotopic composition in terrestrial ecosystems".

Reply: Correct, thanks, we will correct.

[7] L18ff. "in situ" is a vague term. . . destructive as well as non-destructive sampling are always done in situ. On the other hand, the measurements are performed on-line vs offline. Consider using another terminology throughout the MS, e.g. the aforementioned "on-line"

Reply: We disagree. Here a definition of in situ: "A direct measurement of the measurand in its original place". That's exactly what we do, when we measure in the field. If an experiment is conducted in the laboratory and a soil column is prepared, this definition also covers that. Destructive samples are not measured in 'their original place'. In contrast, 'online' is in our opinion a vague term as many people confuse it with being online - so does it mean the data is transmitted online? Here is a definition of on-line: 'On-line means that's it's generally continuous, or constantly taking the measure'. Most in situ studies do not measure continuously.

[8] L19. "disentangle" is used twice in the abstract. Find alternate term here maybe. . .

Reply: We will do so.

[9] L 21. "water stable isotopic compositions".

Reply: See comment above, 'isotope values'

[10] L21-28. Should be one single paragraph.

Reply: Agree, we will change this.

[11] L24. An interface cannot be threefold. You are talking of a continuum here. They are many interfaces present within a continuum and in this particular case three, i.e., the soil-root, soil-atmosphere, and plant-atmosphere interfaces.

Reply: Thank you, we will correct this.

[12] L25. This is a given. Consider omitting.

Reply: OK, we will delete this.

[13] L25-26. "in situ methods for soils are well established". The literature as well as yourself later say otherwise.

Reply: Perhaps 'well established' is the wrong term used. What we mean is that several authors have shown that reliable data can be generated in the field using such methods. We will rephrase this in the revised version.

1. Introduction

[14] L45-48. What is always omitted and should be stated here is the water demand vs water availability. Experts on root water uptake processes know that if a plant does not need to extract water that is "easily" accessible, because it e.g., does not transpire enough or is adapted in this regard. . .it will keep extracting "less available" water. "Easily accessible" water is also poorer in dissolved oxygen and nutrients and might be the last place to look for water for some species. This should be discussed.

Reply: We agree that the uptake of water is by no means a pure question of meeting the transpirative demand of a plant. Therefore, while the relationship between plant water potential and the soil water potential in different depth sets the boundary conditions

for possible uptake depths profiles, there are other factors to consider such as the availability of nutrients, which are commonly distributed exponentially in the soil. We will add a discussion on this aspect as requested.

[15] L52. "at the soil-vegetation-atmosphere interface". See my previous comment.

Reply: We will correct this.

[16] L55-64. Earlier you are talking of water "worlds" but here of "pools". They are not the same. This should be discussed as well.

Reply: See comment above. We will clarify this.

[17] L62-63ff. Edit reference format. It should be "et al."

Reply: We will correct this.

[18] L63-64. "As a consequence, a big question arises: Are all source water studies wrong?": this is a scary statement (!) Also do not say "wrong", as there is a moral aspect to it. "biased" sounds about right.

Reply: We agree and will rephrase this.

1.1 Are ecohydrological source water studies biased? – The need for in situ methods

[19] L70 "low suction tension" :)) Just say low "low tension" or "low soil water tension" Reply: 'suction tension' is the accepted term in soil physics; thus, we will keep it as is. Fig.1. You are missing evaporation! Also phloem-xylem exchange, Péclet effect, and cuticular evaporation could be mentioned. In addition you should use arrows for fluxes rather that for lag time. Anyway, is this (nice) drawing really needed? It is not quite informative, and you don't make use of it in the text.

Reply: Thanks for the comment. It is true that we omitted the abovementioned effects. We still believe that this graphic is a good way to lead the reader to the topic and make them understand why in situ methods can contribute to improve process

understanding. Thus, we would like to keep the graphic. We will add Evaporation and phloem-xylem exchange in the revised version. Peclet effect and cuticular evaporation somehow belong to evaporation; hence, we would add them in brackets () to evaporation.

[20] L79. "water isotope composition values"

Reply: We do not see why 'composition' needs to be added here. See comment above also. We propose to use 'isotope values' throughout the manuscript.

"2 Review: In situ approaches for measuring soil and plant water stable isotope composition"

[21] L102. "2.1 In situ soil water isotope composition depth profiles"

Reply: See comment above.

[22] L103-110. Nice §!

Reply: Thanks!

[23] L105. $\delta$2H needs definition

Reply: We will add this, perhaps as early as in the glossary.

[24] L105-106. Take care of the scientific grammar / isotopic terminology: it should read "The determined isotope composition values agreed well with that of the samples extracted from the soil".

Reply: We will check again for this. However, the term 'isotope composition value' is extremely uncommon. We also did not find it in the terminology papers of Coplen (2011) and Bond & Hobson (2012). The most common and correct according to their definition is using 'delta values', though isotope values are found very often and – in our opinion – not misleading. We will consequently use 'isotope values' or 'delta values'.

[25] L103-212. "The concepts tested in Herbstritt et al. (2012) therefore can be seen

as a baseline for all subsequent in situ soil water isotope studies.": I would not say this at all. Barbara Herbstritt's study is not on soil water, but on meteoritic waters. Their membrane as well as their modus operandi were not further used in the "subsequent in situ soil water isotope studies". This is also the case for Soderberg et al.'s study, which, even though it constitutes the first published work on online soil water vapor measurement, can be seen as an outlier as their method was not further applied. I see that they are two "families" of methods that rely on the same principles: those of Rothfuss et al. (2013) and Volkmann et al. (2014). The first one has seen to date more applications and further developments (Rothfuss et al. 2015; Gangi et al. 2015; Quade et al. 2018; 2019, Oerter et al. 2017, 2019, Oerter and Bowen, 2017, 2019, Kühnhammer et al. 2019) that the second one (Gaj et al. 2016), certainly for the reason that the probes weren't at first off-the-shelf products (although Gaj et al. 2016 used already commercially available soil gas probes). We in Jülich personally did not have the idea to use membranes from Barbara Herbstritt's work, rather from previous applications in soil $CO_2$, $N_2O$, and $CH_4$ sampling [Dinsmore et al., 2009; Hartmann et al., 2011; Neftel et al., 2000]. These studies could be a nice addup to this nice section. Also you are almost up to date: you are missing the following study: Quade, M., et al. (2019). "In-situ Monitoring of Soil Water Isotopic Composition for Partitioning of Evapotranspiration During One Growing Season of Sugar Beet (Beta vulgaris)." Agricultural and Forest Meteorology 266–267: 53–64.

Reply: Thank you very much for this good comment. We will rephrase this and clarify that there are 'two families' originating from different ideas. We also will incorporate a number of L.I. Wassenaar's studies here as we did not correctly acknowledge their early works. We propose to delete the two sentences from l. 115-117 (Thought not strictly measuring. . .baseline for all subsequent in situ soil water isotope studies.) and change L 113: "measurement in soils" to "measurement of soil water". This way, we keep acknowledging their study as it is definitely related for the liquid to vapor phase conversation despite not being a real in situ soil measurement. In regard to the previous studies using $CO_2$, $N_2O$ and $CH_4$ sampling, we decided to add a brief section on the

trigger for one of the families that the in situ methods emerged from. See below for a proposed phrasing. Rothfuss, Vereecken, & Brüggemann (2013) and Rothfuss et al. (2015) tested precision and accuracy of membrane-based in situ approaches in laboratory experiments. Their variety/type of a liquid-vapor equilibrium-based method to measure water stable isotopes emerged from techniques developed to continuously measure soil trace gases (Gut et al. 1998). The membrane material (Accurel PP) was previously already used to investigate vertical $N_2O$ profiles in soils (Neftel et al. 2000), soil $CO_2$, $N_2O$ and $CH_4$ concentrations in response to water table depths (Dinsmore et al 2009) and the effect of drought and fertilization on $CH_4$ concentrations in different soil depths (Hartmann et al 2011). Rothfuss et al. (2013) set up an airtight acrylic vessel filled with fine sand...

[26] L158. "directed in"

Reply: We will change this.

[27] L158-159. Dry air cannot equilibrate with some isotopic composition. Please reformulate

Reply: We will change this.

[28] L169. "combined measurements of $\delta18O$ and $\delta2H$" I presumed.

Reply: Thanks! We will change this.

[29] L213-300. Part 2.2 feels not to fit the scope of the paper, i.e., the "current status of in situ measurements of water stable isotopes in soils and plants". I suggest removing it. It will not distract the reader away from the core of the paper.

Reply: While we agree that the membrane-based soil and xylem based techniques are more recent, we believe that chamber techniques particularly observing transpiration fluxes and isotopes are very crucial in-situ methods that belong in this paper. Also see comments above (main comments).

2.3 In situ measurements of plant xylem water isotopes

[30] L312. "can be observed"

Reply: We will change this.

[31] L312-313, 316ff etc. "xylem water isotope compositions"

Reply: See comments above, compositions is rarely used.

[32] L317-318. Define IRIS and IRMS, and "BSquare-weighted M-regression"

Reply: We will change this and also add to the glossary (the former two)

[33] L320. Use ‰

Reply: We will change this.

[34] L321-349. I am not sure that you should review non-published literature. . .I leave that to the editor to decide.

Reply: See above, meanwhile accepted and online.

[35] L337 (goes as well for at L338). "$\delta$18O xylem isotopic composition values were lower than those of the source water"

Reply: We will change this.

[36] L349. This table is central to your review and should be moved from the SI to the main text and named Table 1! It should be mentioned at the beginning of section 2, so that the readers can consult it while reading.

Reply: Thank you, we agree and will change it to the main text.

[37] L350. "3 Setup, Calibration and Validation of in situ measurements of soil and plant water isotope compositions"

Reply: See above
[38] L354. "for sampling the water vapor"

Reply: We will change this.

[39] L355-358. Exactly my specific comment from before on section 2.2 (+see my general comment)

Reply: We don't see why these methods do not belong into this review. In our opinion the chamber methods are an extremely part of monitoring the plant water cycle. This becomes even more important when speaking about modeling aspects. For example, with measured data on xylem and transpiration water isotopes the development of process-based modeling would benefit extremely – and the reviewer itself suggests to add a modeling part to this review ð§ŸĽ

3.1 Materials and approaches for sampling soil water vapor

[40] L362-363. Only Volkmann and Weiler (2014) built probes in the pure sense. The rest of the authors (apart from Gaj et al., 2016) basically just cut sections of the gas-permeable tubing and took care that it was water tight when installed in the soil and properly connected to the sampling lines.

Reply: Taking a piece of gas permeable membrane and adding capillaries, tubing, etc. is also 'building' or not? And different authors used different approaches for that. Volkmann and Weiler in the end did the same: using semi-permeable material and stick it together. Later, they used probes that were built by a company (Porex). We do not see any major difference here?

[41] L365. "Self-made" is not appropriate. You mean tubing vs membranes inside sensor.

Reply: We will change this.

[42] L367-392. This part echoes my comment from before regarding text at L103-212. I suggest a bit of streamlining and incorporating the one part (L103-212) into the other

(L367-392) or vice versa.

Reply: We will change this.

[43] Two comments here: 1- soil vapor is not necessarily water saturated for a soil liquid-water vapor equilibrium to be reached. Isotopic equilibrium happens between the two phases at relative humidity < 1 at high soil water tension.

Reply: Correct, the statement will be rephrased. In reality, rH almost never reaches 100 percent in soils and still there is isotopic equilibrium. Only under super dry conditions the assumption of equilibrium fractionation should not hold true anymore.

[44] 2- there is no (and should not be!) overpressure inside the tubing, otherwise you push the dry air inside the soil and disturb the vapor-liquid coupled state. For this the sampling lines downstream of vapor collection should be kept as short as possible to minimize resistance to air flow and of course an open-split. But even with long collection lines, the increase of pressure is/should not be observed. You have, on the other hand an overflow (rather than overpressure).

Reply: Thank you, this is correct. We will revise and improve the relevant statements.

[45 ]L380. "access tube just before the analyzer inlet": the technical accepted term is "open-split"

Reply: Another case for the glossary.Thanks!

[46] L385-390. What is the point of having a slightly easier system to implement/use if in the end it might not provide reliable data? My opinion: make a decision as this will influence the readers and potential future users.

Reply: Hm, but you might also ask the other way round: What is the use of a super complicated system that might work under lab conditions, but is completely not feasible in the field and would be limited to one or two research groups with only this expert knowledge? As we state later in the manuscript, one cannot expect a laboratory precision

under field conditions. We have to accept larger uncertainties for field measurements. We believe that constructing a system which is as simple as possible is highly desireable for field application but also generally when we as a community aim at distributing these methods outside of the "isotope experts" community. In this regard, there is a strong advantage of the "pull" method compared to the "push" method. Moreover, in a "pull" system, connections that actually pose the risk of leaks and thereby entering of atmospheric air, can be largely avoided to a degree that the risks gets minimal. We know that this cannot be a universal solution and we will explicitly give examples for both systems and where they are appropriate.

[47] L390-391. You said this already above. Consider erasing. Reply: We will delete the statement.

3.2 Saturation of water vapor, condensation and dilution

[48] L400-410. A bit of reorganizing would help: you need to distinguish between condensation prior sampling starts from condensation during measurements. To get rid of the first problem, you should do iii) and to avoid the second kind of problem you should do points i) and ii).

Reply: Thank you for this comment. This is a very good point and we will clarify that in the revised version.

[49] Ideally you need (i) a three way valve after collection point and before dilution and (ii) a two way "normally open" valve between dilution point and open-split. This former allows you to: 1) flush the liquid water out of the gas-permeable tubing and out of the downstream sampling line prior measurement [valve is turned on and act as open-split] 2) flush the content of the gas-permeable tubing to the laser- spectrometer [valve is turned off] 3) flush the sampling line after dilution point and to the laser spectrometer open-split once sampling is complete to lower the water vapor mixing ratio in it and avoid condensation problems in between measurement phases [valve is turned off and act as open-split]. Valve (ii) is then closed to keep the sampling line dry and the laser

spectrometer samples from its own open-split. Finally for increased security, you shall heat the sampling lines from the soil/trunk surface to dilution point.

Reply: Thank you, this is a good suggestion as well. We will re-organize this section and incorporate the above statements.

[50] This could be incorporated into Figure 7. Reply: True. However, then this already fairly complex graphic will become even more complex. This would be no problem for people who have done such measurements already (such as the reviewer and us) but might irritate people who might apply in situ methods in the future. We propose to include it in the text but omit in the graphic for simplicity.

[51] L420. "Correction for water vapor concentration dependency of the laser spectrometer isotopic composition raw readings"

Reply: We will correct this.

[52] L426. "3.3.1 Soil water vapor isotope standards"

Reply: We will correct this.

[53] L427. Not only with the same medium but thanks to the exact same method: the standard water vapor has to be collected in the same manner as that of the soil, i.e.: (i) pass through the walls of the same membrane product coming from the same production batch (e.g., tubing of the same age, thus having the same permeability to water vapor), be exposed to equivalent environment (pH, chemical conductivity of the soil solution). (ii) at an equivalent flow rate as for the samples Also not to forget: a soil is not only characterized by its natural texture, but also structure, so the soil water vapor standards should ideally have the same structure on top of the same texture as the investigated soil. In relation to this: in my group we however did not see an effect of soil water content on raw isotopic readings. In Quade et al. (2018), we argue that what Oerter et al. (2017) see is due – at least partly – to the fact that they test the membrane on unstructured soils.

Reply: It is correct what you state. We will include these aspects into the relevant section.

[54] L453-459. You should mention of course the work of Markus Schmidt and colleagues, however do not display their equations as they are specific to their instrument. Also they used one of the very first L1102i (number 8 if I remember correctly :) and Picarro[®] (and this goes for other companies) have since greatly reduced the linear response of their instruments to water vapor activity. What you should write, on the other hand, is a short description on how these dependencies should be investigated using, e.g. the aforementioned soil water vapor isotopic standards at different dilution rates.

Reply: Correct, this is a good suggestion. We will briefly describe this in the revised manuscript.

3.3.3 Other corrections (mineral mediated fractionation, organic contamination, carrier gas and biogenic matrix effects)

[55] L469-470. This relates to artificial unstructured soil mixtures. . .see my previous comment. We are dealing with. . .soils, so structure (bulk density for instance) might play a significant role!

Reply: OK, see above, we will elaborate on that.

[56] L475-487. If this apply to soil and plant water online measurements, the associated effect on raw isotopic composition readings should be orders of magnitude lower than for extracted water after inverse sublimation of for evaporated water following pyrolysis. . . You could mention that this has not been investigated but you don't need to review the literature that deals with VOC contamination on liquid samples (out of scope). In addition: what about the material out of which the sampling lines are made of and about their isotope effects? (there is also a dynamic adsorption-desorption equilibrium happening in them) Which one should we take - plastic or metal? Which kind of plastic, which kind of metal??

Reply: Good point as well. We will add a sentence on recommended material based on the existing studies.

3.3.4 Drift correction

[57] L494. Split "Conversion of vapor to liquid values" from "3.3.4 Drift correction". These are unrelated things.

Reply: We will do that in the revised version.

[58] L497-498. You can omit the book of Clark and Fritz (this goes as well for at L122): the (peer-reviewed) works of the great Juske Horita and Mustafa Majoube are enough :) These equations refer then to another state of equilibrium, namely "static equilibrium". What we deal with here during purping is dynamic quasi isotopic-equilibrium. Rothfuss et al. (2013) showed that the purging and/or the PP material had an isotopic effect for 1H2H16O. And it is up to now the only real calibration available. These equations should be referred over those of JH and MM or to the very least mentioned.

Reply: Thank you, you are right. We will: i) Take out the Clark and Fritz citation; ii) incorporate the suggestions and equations from Rothfuss et al. (2013) into the revised version.

3.4 Validation –comparing apples and pears?

[59] L538. Oerter et al. (2017) does not "propose a novel, innovative way of calibrating in situ data of soil water isotopes". You refer to Oerter and Bowen (2019), certainly.

Reply: Actually, no. The first time they propose this was in the 2017 papers. They only added the carrier gas effects in 2019 based on Gralher et al. 2016. We will add the latter publication and elaborate on these corrections in the relevant section.

[60] L545-560. There is a good chance that such regression analyses will add more uncertainty to the measurements when not performed carefully. Points (i)-(ii) omit the importance of the soil structure on soil water vapor flow. A note: the addition of water

to a dry soil does not ensure a homogeneous soil water content. Points (v)-(vi) apply to deconstructed soil samples, see previous comments

Reply: Yes, but same applies to any measurement that is not carried out carefully. . .we will try to emphasize that more in the revised version.

4 Summary and Outlook

[61] L586-587. "we spare modelling approaches in this review, but refer to recent developments,. . .": see my general comment.

Reply: ..and our answer above.

[62] L618-621. In my opinion, this is not the way to go. It is much better to prepare different standards in an appropriate manner (repacked at the same field density) reflecting the different textures encountered in the field, so accounting for e.g. the effect of hydration and isotopic fractionation around clay particles. It has to be proven if, indeed SWC plays a role when the soil is structured: my group has never seen such an effect (see Rothfuss et al. 2013, 2015; Quade et al. 2018; Kühnhammer et al. 2019) with compacted soil: this should mentioned/discussed. L649-653. Rothfuss et al. (2013) did this. . .also do not cite a paper under review (Marshall et al. 2019).

Reply: We will add further explanation to that section. At least the effect of different grain sizes can be accounted for when preparing standards using field soil easily. That leaves the two aspects of SWC and packing density (or bulk density). We will discuss this in the revised version. However, consider our previous comments in regard to accuracy vs. practicability. We will also try to touch on this in the revised version.

[63] L664-661. This is a very good idea! However the terminology used in a bit funky: the relative humidity is the ratio of actual to saturated water vapor pressures (not water vapor concentration) in the original definition. Also you certainly mean water vapor dry mixing ratio, which is typically expressed in vol-ppm. Why should the relative humidity be higher than 0.9 specifically? Look at the Kelvin equation and you'll see that for soil

pF=7, relative humidity should yield to 0.97.

Reply: Thank you, we will rephrase to make this section more concise and clear.

[64] L671. Define compartments

Reply : We will do so in the revised manuscript.

[65] L674. This only applies to trees, right..?

Reply: Please clarify? The section mentions how both soil and tree approaches should be combined, so we are missing the point here?

[66] L689-693. Why would you say that? Scientist from other field might say otherwise. . .you can say, on the other hand, that online non-destructive methods are more complex to install/implement than destructive sampling ones. But as long as we do not make complex things complicated, we are on the safe side :)!

Reply: Correct, it is a matter of standpoint/experience and knowledge in the end. However, it is an extreme effort and hassle up to now to obtain good in situ measurements in the field. If you are starting with it, it is overwhelming. What we try to communicate here is that one should start from easy and then add up complexity instead of starting with the most complex system that works in the lab but is logistically not feasible in the field. In this regard, I speak out of experience because it happened in our group and if this is the way to go it will scare people from trying in situ methods. In our case, the attempt to start with the most complex system in the field resulted in the need for daily supervision of the instruments. In the end cost (labor and general effort) and benefit (good measurements) need to be in balance. My [MB] personal opinion is that for most applications it is possible to use the pull-only mode with a minimum amount of connections (ideally: 1-2 maximum) to obtain reliable results. So, indeed, why make complex things complicated! ð§ŸĽ
* * *
[Figure]

600, 2019.

---

## Author Comment (AC3) · 28 Mar 2020

We kindly thank reviewer#2 for this very positive review of our manuscript. We will try to further improve it during the revision process and appreciate the additional (technical) comments and suggestions.

[1] The title is slightly misleading however, as the manuscript barely deals with the "two or x water worlds" hypothesis. I suggest to remove all parts before and including the ":" from the title and just call it "A review and future perspective on in situ measurements

of water stable isotopes in soils and plants".

Reply: Thank you. Please also refer to our reply to reviewer#1. It is true that we do not refer extensively to the two water worlds hypothesis. However, the title 'X water worlds' was meant to relate to it in a slightly provocative way. We believe that is it is important to clarify that this hypothesis which has certainly had impact on the community has been seen critically by large parts of the (European) community. It has been shown that it can be explained by simple soil hydraulic relationships that are used by soil scientists since decades. However, the hypothesis has had a very positive impact on the advancement of process understanding. The in situ methods fall exactly into this category. We believe that for a review paper it is important to put those aspects into a context and the title and Fig.1 are meant to expose that soil-plant water relationships are more complex than the hypothesis states. However, as both reviewers suggested to change the title, we kindly ask for a comment of the editor. As stated in the reply to reviewer#1 we would like to keep as is because of i) put the in situ measurements into a larger context and ii) approach a wider community to thing further than 'two water worlds'. Perhaps a few text-additions might also help to clarify that.

I only have some minor comments for clarification or little bit more detail and suggestions for typos. Specific comments

[2] L38: the more easily accessible water would be soil water that eventually becomes stream water, currently the sentence sounds like stream water is directly sampled

Reply: Thank you, we will rephrase and clarify.

[3] L95: unclear if "leaf chamber" is a physical e.g. plastic chamber where leaves are studied or if stomatal openings, the sites of transpiration, are meant.

Reply: We refer to an actual physical chamber and will define this properly.

[4] L120: where's the difference of fractionation factors and equilibrium fractionation factors? Be a bite more specific what is exactly meant with the first term

Reply: Thank you. We will rephrase and clarify.

[5] L164: which other value was used to linearly correct the measured hydrogen isotope values in vapor in the study of Rothfuass et al. 2013? Further, how did Rothfuss et al (2015) proof that their isotope values were reliable? They must have compared it to something.

Reply: For the conversion of vapor to liquid values, Rothfuss et al. (2013) use only temperature. Rothfuss et al. (2015) are not completely clear on to what they compared their values to. We believe they compared the obtained values to the ones from water that was used to saturate their soil column. We will add this information in the revised version.

[6] L173: according to Line 128, the first in-situ study in semiarid areas was by Sodeberg et al (2012) and not Gaj et al. (2016). Please clarify.

Reply: This is true. They were the first even though it did not work very well. We will correct that in a revised version.

[7] L183: incomplete soil water extraction by which system?

Reply: We will add the missing information, the authors used a cryogenic vacuum extraction system.

[8] L200: since this system is the most complete, I am lacking a bit of information of what this system consists of exactly. Especially since later the authors of this manuscript suggest a more general, expanded approach to the cited study. The other methods were well described but here the description is a bit lacking. It seems to be a vapor-permeable membrane technique, but the advantage or difference to previously described methods is unclear.

Reply: Thank you. Indeed, the description of this section is not completely comprehensive. We will provide a better summary of the study in the revised manuscript.

[9] L210: I would not call this "trueness" but "accuracy". Since there is also a discussion ongoing of which water is exactly sampled by each method and which method can give "correct/true" values under which conditions.

Reply: I agree, but we used the phrase that was used in the cited publication and would stick with this to not confuse the readers.

[10] L408: method number 3 does not sound like a system to prevent condensation. The condensate between each flushing is just removed.

Reply: Correct, thanks. We will clarify.

[11] Also for the example of Figure 5, where is the data coming from? Did you conduct this experiment or is it from another study?

[12] Reply: This is our own data. We will add a brief explanation to clarify that.

[13] L445: the meaning of the sentence becomes unclear starting with "correction has mostly been applied [. . .]"

Reply: We will rephrase this part.

[14] L488: The description of carrier gas effects is very short compared to the rest

Reply: That is correct. In the revised version of the manuscript, we will extend that section.

[15] L545: does the proposed method have to be repeated for each individual study? I think that the final multiple regression results should be general and applicable for many studies. Please elaborate on this.

Reply: Ok, we will do so.

[16] L579: I do not see how the example of mobile water is related to the sentence before, that in situ methods overcome current method limitations.

Reply: The statement is meant in a way that by using destructive sampling and extraction methods, we do not really know which pool we extract (only the mobile water? Mobile plus a bit of tighter bond water? All water?). For example, using 105°C for extraction will extract different pools for sand, silt, or clay soils. With the in situ methods one always measures the mobile part.

Technical comments

[17] L61: citation: Hendry et al. 2015. As changes in citation style appears again in the manuscript, please keep it uniform everywhere.

Reply: Thank you, we will keep this consistent.

[18] L83: sentence "and this is where. . ." should be shifted, as it currently reads very confusing. For example, "The scientific community agrees that one of the most important steps to investigate, disentangle, quantify and incorporate. . .."

Reply: Thank you, we will reorganize the sentence.

[19] L87: delete the one "a" before "the same"

Reply: We will do so.

[20] L91: "kill the plant" please correct

Reply: We will do so.

[21] L92: Sentence sounds like time and costs of destructibe sampling result in larger sample amounts.

Reply: We will rephrase.

[22] L94: delete "of" L169: "O" for oxygen missing at the end of the sentence with "delta 18". Also unclear what combined measurement is referred to? I assume d18O in water and carbondioxide?

Reply: This sentence is very confusing. We will improve that in the revised manuscript.

[23] L188:delete either "of" or "for" L200: "apllied" please correct

Reply: We will do so.

[24] L213: write already "isotopes" in the heading for 2.2

Reply: We will do so.

[25] L252: "Gordon" missing

Reply: We will correct this. Thank you.

[26] L304: delete "and" L375: delete one "now"

Reply: We will correct this. Thank you.

[27] L415: "disappears" spelling

Reply: We will correct this. Thank you.

[28] L454: use "positive correlation" instead of "positive effect", as this sounds beneficial

Reply: We will correct this. Thank you.

[29] L465: "no" should be "not" Reply: We will correct this. Thank you.

[30] L472: "A" should be "As [stated above]" Reply: We will correct this. Thank you.

[31] L512: "is" should be "as" Reply: We will correct this. Thank you.

[32] L514: do you mean "previous" instead of "succeeding"?

Reply: Thank you! You are right! We will correct this.

[33] L671: Fig 7 instead of Fig 6

Reply: We will correct this. Thank you.

600, 2019.

---

## Author Response (AR1)

Dear Christine (Editor),

Please find attached our revised version of the manuscript **hess-2019-600: X Water Worlds and how to investigate them: A review and future perspective on in situ measurements of water stable isotopes in soils and plants.**

We invested substantial efforts in including the many useful and constructive comments provided by Youri Rothfuss (Rev.#1), an anonymous reviewer (Rev.#2) and Erik Oerter (short comment). We would like to thank them once again in helping to improve this manuscript. All changes made are in accordance to our responses to the reviewers which we provided earlier.

The most prominent changes we summarize as follows:

- Changed the manuscript title (New: 'In situ measurements of soil and plant water isotopes: A review of approaches, practical considerations and a vision for the future');
- Included new chapter on physically based isotope-enabled modeling (chapter 4 in new version) and a corresponding table (table 2); this section was prepared by a new co-author (K. Kühnhammer)
- Added Glossary defining the main terms used throughout the review
- Revised Figure 1 (minor changes)
- Updated literature to current (2020), checked for consistency in isotope terminology

We believe that the revised version provides a thorough overview on the status quo and current forefronts of in situ measurements and their application. We look forward to hear your opinion,

Kind regards,

Kathrin Kühnhammer, Maren Dubbert and Matthias Beyer

---

## Author Response (AR2)

Thank you for accepting our manuscript. We appreciate all the comments and good suggestions during the review process. We hope to have made a useful contribution that helps the community. We only made changes in accordance to Reviewer #1's suggestions in this version and added the acknowledgements at the end of the main text.

Thanks, and kind regards,

Matthias Beyer in the name of the authors.

**Response to Reviewer#1 Y.Rothfuss:**

First, sorry for not providing a point-to-point reply. We thought as this has been done as response to the first-round comments it is not required again as the changes that will be made have been indicated there already.

We agree to what the review Y. Rothfuss states in terms of terminology: 'Isotope [or isotopic] composition [or signature]" is, to the contrary of what the authors say, the accepted terminology'. This is true and our statement that this notation is rather uncommon was not correct. We changed the manuscript and use 'isotopic composition' throughout the text. For expressing a specific value, however, we do not follow the reviewers' suggestion to use 'water stable isotopic composition value' – though the explanation given by Y. Rothfuss might be correct, the term is simply an unnecessary and lengthy construct. In fact, the reviewer does not even use it in their own publications. After some more research (not to prove anything rather to learn what terminology is most commonly used) we conclude that using '$\delta$ value' is the most commonly used phrase when referring to a specific value. (we cross-checked a couple of authors: G. Bowen: isotope values; T. Volkmann: $\delta$ value/ sample value; M. Sprenger: $\delta$ or isotope value; K. Kühnhammer: $\delta$ value; Y. Rothfuss: $\delta$ value or value of $\delta$). We hope that this is OK now. We eliminated the term 'isotope value' from the manuscript.